# UAS Chromatograph for Atmospheric Trace Species (UCATS) – a versatile instrument for trace gas measurements on airborne platforms

Eric J. Hintsa,[1,2] Fred L. Moore,[1,2] Dale F. Hurst,[1,2] Geoff S. Dutton,[1,2] Bradley D. Hall,[2] J. David Nance,[1,2] Ben R. Miller,[1,2] Stephen A. Montzka,[2] Laura P. Wolton,[1,2] Audra McClure-Begley,[1,2] James W. Elkins,[2] Emrys G. Hall,[1,2] Allen F. Jordan,[1,2] Andrew W. Rollins,[3] Troy D. Thornberry,[1,3] Laurel A. Watts,[1,3] Chelsea R. Thompson,[1,3] Jeff Peischl,[1,3] Ilann Bourgeois,[1,3] Thomas B. Ryerson,[3] Bruce C. Daube,[4] Yenny Gonzalez Ramos,[4,5,6] Roisin Commane,[4,7,8] Gregory W. Santoni,[4] Jasna V. Pittman,[4] Steven C. Wofsy,[4] Eric Kort,[9] Glenn S. Diskin,[10] and T. Paul Bui[11]

[1]Cooperative Institute for Research in Environmental Sciences, University of Colorado, Boulder, 80309, U.S.A.
[2]Global Monitoring Laboratory, NOAA, Boulder, CO 80305, U.S.A.
[3]Chemical Sciences Laboratory, NOAA, Boulder, CO 80305, U.S.A.
[4]John A. Paulson School of Engineering and Applied Sciences, Harvard University, Cambridge, MA 02138, U.S.A.
[5]CIMEL Electronique, Paris, 75011, France.
[6]Izaña Atmospheric Research Centre, Santa Cruz de Tenerife, 38001, Spain.
[7]Dept. of Earth and Environmental Science, Columbia University, New York, NY 10027, U.S.A.
[8]Lamont-Doherty Earth Observatory, Columbia University, Palisades, NY 10964, U.S.A.
[9]Climate and Space Sciences and Engineering, University of Michigan, Ann Arbor, MI 48109
[10]NASA Langley Research Center, Hampton, VA 23681, U.S.A.
[11]NASA Ames Research Center, Mountain View, CA 94035, U.S.A.

*Correspondence to*: Eric Hintsa (Eric.J.Hintsa@noaa.gov)

**Abstract.** UCATS (the UAS Chromatograph for Atmospheric Trace Species) was designed and built for observations of important atmospheric trace gases from unmanned aircraft systems (UAS) in the upper troposphere and lower stratosphere (UTLS). Initially it measured major chlorofluorocarbons (CFCs) and the stratospheric transport tracers nitrous oxide ($N_2O$) and sulfur hexafluoride ($SF_6$), using gas chromatography with electron capture detection. Compact commercial absorption spectrometers for ozone ($O_3$) and water vapor ($H_2O$) were added to enhance its capabilities on platforms with relatively small payloads. UCATS has since been reconfigured to measure methane ($CH_4$), carbon monoxide (CO), and molecular hydrogen ($H_2$) instead of CFCs and has undergone numerous upgrades to its subsystems. It has served as part of large payloads on stratospheric UAS missions to probe the tropical tropopause region and transport of air into the stratosphere, in piloted aircraft studies of greenhouse gases, transport, and chemistry in the troposphere, and in 2021 is scheduled to return to the study of stratospheric ozone and halogen compounds, one of its original goals. Each deployment brought different challenges, which were largely met or resolved. The design, capabilities, modifications and some results from UCATS are shown and described here, including changes for future missions.

## 1 Introduction

Accurate and precise measurements of trace gases and other atmospheric parameters have resulted in an ever more detailed understanding of the chemistry and physics of Earth's atmosphere. This has allowed progress on environmental issues of global concern, including stratospheric ozone depletion and air pollution in the lower atmosphere. For example, after the first report of the Antarctic "ozone hole" [Farman et al., 1985], a combination of measurements from balloons, aircraft, and satellites, backed by a wide range of laboratory, theoretical and modeling studies, allowed a sufficient grasp of the problem to develop an effective international response relatively quickly [Douglass et al., 2014]. Though ozone loss and organic halogen emissions are still ongoing matters of concern, climate change driven by greenhouse gas emissions is now the overarching environmental problem today, while air quality continues to be an important issue as well.

The Halocarbons and other Atmospheric Trace Species (HATS) group in what is now the Global Monitoring Laboratory (GML) at the National Oceanic and Atmospheric Administration (NOAA) in Boulder, CO has long been involved in measuring $N_2O$, CFCs and other trace gases, primarily by using gas chromatography (GC) with electron capture detectors (ECDs). This led to participation in a series of airborne missions to study halogen budgets, ozone loss, and stratospheric transport, starting on the NASA ER-2 aircraft in 1991 [Elkins et al., 1996]. GCs are well-suited to measuring multiple trace species in the atmosphere, because with an appropriate column, several compounds can be separated and detected with the same instrument. ECDs are extremely sensitive (a few ppt or better) to halogen-containing compounds, including ozone depleting substances (ODS), and with appropriate modifications can be used to detect other molecules as well. A few other GC instruments have been used for in situ measurements of CFCs [Tyson et al., 1978; Vedder et al., 1983], and they can now be coupled with mass spectrometric detection [e.g., Apel et al., 2015] as well. With the advent of unmanned aircraft systems (UAS), the potential emerged to extend scientific airborne missions to longer durations and other experiments that were not possible with manned aircraft, as well as eliminate some of the danger of flying piloted aircraft in remote regions. Accordingly, the UAS Chromatograph for Atmospheric Trace Species (UCATS), a smaller and lighter weight version of previous aircraft instruments, was designed and built to measure ODS and other trace gases on UAS missions. These began in 2005 with the Altair UAS, a high altitude version of the General Atomics Predator B. Given the limited payload capacity of Altair, small and lightweight ozone and water vapor sensors were installed inside UCATS to generate a more complete data set. After two missions on Altair, UCATS joined the payload of the National Science Foundation/National Center for Atmospheric Research (NSF/NCAR) Gulfstream-V (GV) for the START-08 (Stratosphere-Troposphere Analyses of Regional Transport; 2008) and HIPPO (HIAPER Pole-to-Pole Observations; 2009-11) missions [Pan et al., 2010; Wofsy, 2011], which included vertical profile measurements from near the surface to the lower stratosphere. In 2010, UCATS flew on the NASA Global Hawk UAS for the Global Hawk Pacific (GloPac) demonstration project, and participated in the Airborne Tropical Tropopause Experiment (ATTREX; Jensen et al., 2013) from 2011 to 2014, to study dehydration, transport, and ozone chemistry in the tropical tropopause layer (TTL). Most recently, UCATS completed the Atmospheric Tomography (ATom; 2016-18) mission, for which the NASA DC-8 aircraft sampled the remote atmosphere over the Atlantic, Pacific and Southern Oceans and parts of the Arctic and Antarctic from near the

surface to above 12 km in different seasons.  Many of these missions required changes to UCATS, and components were also upgraded when possible.  The end result is a compact instrument for UAS and piloted aircraft, capable of measurements of atmospheric composition, chemistry, and transport in the stratosphere and troposphere.  We describe the design and components of UCATS in Sect. 2, focusing on ATom, the most recent mission for UCATS; improvements and modifications over the course of its missions in Sect. 3, and data and intercomparisons from some of the field campaigns in Sect. 4, with a short summary including future plans in Sect. 5.

## 2 Instrument design

At its core, UCATS is similar to previous gas chromatograph (GC) instruments designed and built for aircraft and balloon platforms [Elkins et al., 1996; Romashkin et al., 2001; Moore et al., 2003].  It combines a two-channel GC that is a lighter and much more compact version of the four-channel ACATS-IV (Airborne Chromatography for Atmospheric Trace Species IV; Elkins et al., 1996) instrument and incorporates some of the advances in fast chromatography from the LACE (Lightweight Airborne Chromatograph Experiment; Moore et al., 2003) instrument for balloons, along with small ozone and water vapor sensors.  Fig. 1 shows a block diagram of UCATS with all the major internal components; more detailed drawings of the ozone and water instruments are included in the appendix (Figs. A1a and A1b).  Each part of UCATS is described in the following subsections.

### 2.1 Gas chromatographs

Both GC channels use Valco 10- and 12-port 2-position valves (VICI, Houston, TX) to control flow switching, and ECDs (Valco and Shimadzu) to detect specific trace species with high precision, with added "dopant" gas as needed [Phillips et al., 1979; Fehsenfeld et al., 1981].  In its original configuration, one channel used OV-101 in packed columns to separate and measure CFC-12, Halon-1211, and CFC-11 every 70 seconds, similar to the Lightweight Airborne Chromatograph Experiment (LACE) [Moore et al., 2003].  After the initial Altair flights in 2005, these were replaced with Unibeads (pre-column, ~1 m length) and molecular sieve 5A (main column, 0.7 M) to measure molecular hydrogen ($H_2$), methane ($CH_4$), and carbon monoxide (CO) every 140 seconds.  A tiny flow of nitrous oxide ($N_2O$) dopant (~.003 sccm) added to the ECD is required for adequate sensitivity.  Flows and column temperatures varied over different missions; in ATom, with the chromatography optimized for both the troposphere and stratosphere, at a temperature of 94 °C the $N_2$ carrier gas flow was 60 sccm, with 4 seconds of "pre-emphasis" at 100 sccm at the start of the injection, to rapidly bring the pre-column up to the same pressure as the main column.  The pre-column was back-flushed after 25 seconds to remove any remaining compounds over the remainder of the 140 second time window.

The second channel uses a pre-column (0.6 m) and main column (1.8 m) of Porapak Q, followed by a post-column of 5A molecular sieve (originally 0.20 m, now 0.25 m), to measure sulfur hexafluoride ($SF_6$) and $N_2O$ every 70 seconds; doping the nitrogen carrier gas with $CO_2$ enhances the ECD response to $N_2O$.  The pre- and main columns were maintained at 91 °C, and the post-column at ~120 °C (for the shorter version) and 190 °C (longer version; changed in 2011).  Carrier gas flows in ATom were 55 sccm, with the flow in the pre-column reversed after 13-14 seconds.  Since the backflush switches occur early in the cycle, there is sufficient time for the pre-columns in each channel to be cleaned out, even with lower flow rates compared to the main flows.  All the columns used were packed in 3.2 mm O.D. stainless steel tubing, wound around a circular mandrill with heater cartridges and an RTD to control temperature, and packaged in insulated metal cans.  The ECDs were packaged in similar cans, but sealed and supplied with a ~5 sccm "purge" flow to prevent oxidation, and maintained at 330-350 °C.

Chromatograms are similar to those in Moore et al. [2003], Figures 7c and 9.  ECDs provide very high sensitivity (in the low part-per-trillion, ppt, range) but can have non-linearity, particularly for doped channels, where secondary ion-molecule reactions are used to detect trace species.  UCATS was calibrated on the ground during each mission, with a set of standards spanning the range of expected atmospheric concentrations (typically 30-100% of those in background tropospheric air), and occasionally including zero air to check baselines.  From these experiments, calibration curves for each molecule are calculated, including an estimate of the error in the calibration.  An example of a calibration curve for $N_2O$ is included in the

Appendix (Figure A2); all other curves were even closer to linear. In flight, a calibration standard from
compressed background tropospheric air is injected every 6-10 minutes, and the peak heights of air samples
and standards are analyzed with the calibration curves to generate a time series of mixing ratios for each
molecule in sampled air.

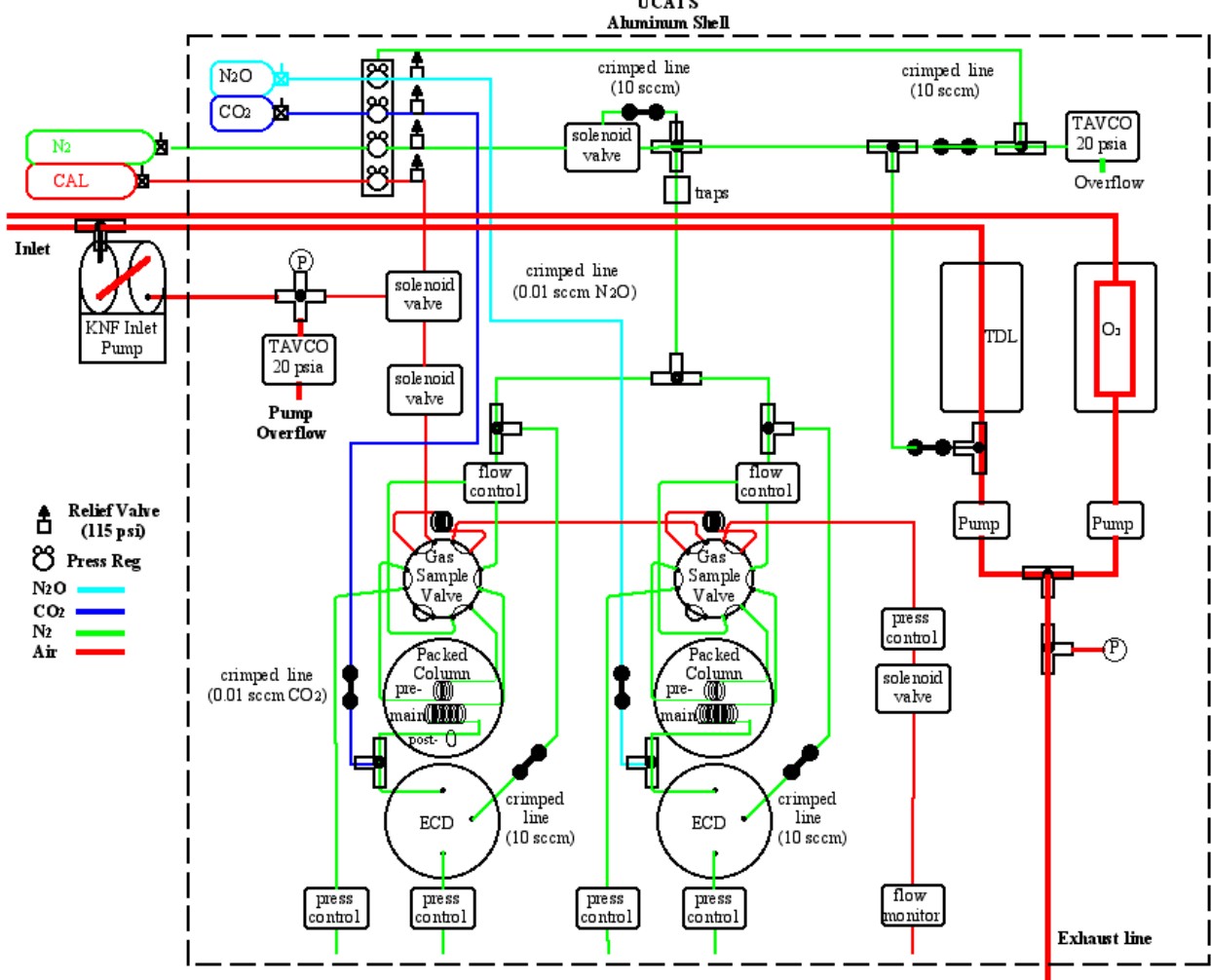

**Figure 1: Schematic of UCATS as flown in the ATom mission showing all major components. Red lines indicate the**
**flow of ambient air through the instrument and blue lines indicate ECD dopant flow. All regulators ("Press Reg")**
**are single-stage, and kept at a constant external pressure by a small flow of carrier gas and a Tavco absolute**
**pressure controller (green line, top) to improve stability. Pressure is measured at points in the system marked**
**"P", as well as at regulators and controllers. Green "crimped lines" typically provide 5-10 $cm^3$/min purge flows to**
**keep the ECDs and TDL cell clean and dry when the instrument is powered off, "make-up" flows to the ECDs when**
**operating, and flows to pressurize the regulators. $N_2$ carrier gas (green) is purified through a set of molecular**
**sieve, Hopcalite, and activated charcoal traps, and a hot zirconium getter, labeled as "traps" in the figure, before**
**being sent to the GCs. Bottles for $N_2$ and calibrated air were located externally for ATom and prior missions; they**
**will be mounted inside the UCATS shell for the DCOTSS mission in 2021. Internal details of the ozone and water**
**instruments are shown in Figures A1a and A1b.**

## 2.2 Air flows and sampling

Ambient air is drawn into UCATS from a side-facing or rear-facing inlet extending 25-30 cm from the skin of the aircraft (outside the aircraft boundary layer) through stainless steel and Synflex tubing; sample flow tubing inside the GC is stainless steel. Air is pressurized in the GC sample loops by an external two-stage KNF diaphragm pump (Model UN726, with Teflon-coated heads and diaphragms) and maintained at 1225 hPa with an absolute pressure relief valve (Tavco, Inc.; Chatsworth, CA); excess air is dumped through the Tavco overflow. Air flows at approximately 80 sccm sequentially through the two sample loops (~0.5 cc volume, one for each channel) and a flow meter, and is controlled by solenoid valves and a pressure regulator set at 1080 hPa on the outlet. Every 70 or 140 seconds the contents of the sample loops are injected by the 2-position Valco valves onto the pre-columns, providing a discontinuous ~2 second "snapshot" measurement of ambient air.

## 2.3 Water vapor

The sample air flow is split just upstream of the GC pump to feed a tunable diode laser (TDL) hygrometer with its own pump (KNF, model NMP850) downstream of the absorption cell. The original hygrometer, a custom commercial sensor from Maycomm, Inc., used infrared absorption at 1.37 μm with second harmonic detection to measure water vapor. Since water vapor number densities span 5 orders of magnitude from the surface to the stratosphere, the laser beam was split into two optical paths, a 13.4 cm "short path" for measurements from the surface to the mid-troposphere (40,000 to 500 parts per million [ppm]), and a 403 cm multi-pass "long path" for measurements from the mid-troposphere into the stratosphere (1000 to <5 ppm). On the Altair missions, with a minimal payload, a small Vaisala probe was installed on the inlet for measurements of temperature, pressure, and relative humidity. This was not used subsequently, as the payloads on larger aircraft included dedicated instruments for meteorological measurements.

During ATom, the TDL hygrometer in UCATS was upgraded with a new model from Port City Instruments (Reno, NV), the successor to Maycomm. It is similar in concept and uses a distributed feedback laser (DFB) to scan across two closely spaced absorption lines near 2.574 μm. Absorption at this wavelength is much stronger than in the original instrument, allowing higher sensitivity in the stratosphere. As before, the laser beam is split into two optical paths, with the short path (5.14 cm) for high values of tropospheric water (~2000-40000 ppm) using direct absorption. The long path (280.0 cm) is used with second harmonic detection for water vapor from 0-100 ppm, and intermediate values (100-5000 ppm) are measured using the long path with direct absorption. A second weak absorption line is also analyzed with direct absorption for water vapor mixing ratios above 1000 ppm; this is not being used at present. Both long and short path spectra are recorded simultaneously, with each scan taking approximately 200 msec. All four measurements of water vapor are calculated, then each is averaged together for ~1 Hz output on a serial data line. All data are recorded by the UCATS computer and the appropriate value for display and archiving is chosen based on the range of pressure and water vapor. Both instruments required extensive calibration using prepared water vapor standards and frost point hygrometers for accurate measurements. The new instrument allows higher precision (±0.1 ppm or better) measurements of water vapor in the stratosphere compared to the original instrument, which was limited to ±1 ppm.

## 2.4 Ozone

Ozone was measured by direct absorption (Beer-Lambert Law) UV photometers from 2B Technologies (Boulder, CO), modified for high altitude operation and mounted inside the UCATS package. The initial ozone instrument was a 2B Model 205; modifications included a stronger pump (KNF, model UNMP-830), a small metal cylinder upstream of the pump to dampen pressure fluctuations that could degrade the measurement precision, $O_3$ scrubbers with manganese dioxide ($MnO_2$)-coated screens (Thermo Fisher), and pressure sensors with a range from 0 to over 1000 hPa (Honeywell ASDX series). Ambient air was brought to the ozone instruments from the inlet through a separate Teflon tube (6.35 mm O.D.), with the exhaust from the ozone and water instruments combined inside UCATS and released outside the aircraft. The Model 205 is a dual-beam photometer, with the flow continuously split between unscrubbed (ambient) air into one cell and scrubbed (ozone-free) air into the other. Two photodiodes located at the end of 15-cm long absorption cells

measure the intensity of 254 nm radiation emitted from a mercury lamp.  Ozone concentrations are calculated from the ratio of measured intensities through the cells with scrubbed and unscrubbed air according to the Beer-Lambert Law.  The flow paths are switched by solenoids every 2 seconds, to allow alternating measurements of ambient and scrubbed air in each cell, with data averaged to 10 seconds on the original model.  The instruments are checked against a NIST-traceable calibration system (Thermo Fisher, Model 49i) on the ground before and after every mission.  For ATom, a new and more sensitive 2B Model 211 ozone photometer was added to UCATS in addition to the original Model 205, with similar modifications as before, and additional changes as described in Sect. 3.

**2.5 Physical characteristics**

The overall dimensions of UCATS were initially 41 x 46 x 26 cm, with a weight of 29 kg.  To integrate the new water and ozone instruments for ATom, an additional section was added to the top, increasing the height from 26 to 33 cm, and the weight to 33 kg.  The external GC pump weighs an additional 5 kg, and fiber-wrapped aluminum bottles (SCI Composites; now Worthington Industries) for compressed nitrogen ($N_2$) carrier gas (model 687) and dry, whole air calibration gas (model 209) for the GC, both filled to ~13000 kPa, together weigh approximately 7 kg.  The total $N_2$ flow (carrier gas, backflush, purge flows) is about 300 sccm. The $N_2$ bottle needed to be filled every one or two flights, the calibration gas was filled once per deployment, and the small dopant bottles could last for over a year without refilling.  For flights on passenger aircraft, such as the DC-8 for ATom, the $N_2$ and air bottles can be replaced by larger gas cylinders as weight and space allow. UCATS is powered by 28 V DC, and the complete package draws 12 A at startup (~350 W), decreasing to 150 W after the heaters warm up (~30 minutes).  The majority of the power is consumed by the column and ECD heaters; the GC pump and the TDL use about 1 A each, ozone less than 0.5 A, and other electronics about 1-2 A.  Different voltages (+5, ±12, 15, and 24 V) are supplied by Vicor DC-DC converters.  The internal wiring in the ECD cans are carefully adjusted to minimize electrical noise on the detector circuits; no other electromagnetic compatibility issues were observed.  UCATS is controlled by an Ampro computer with the QNX operating system, and data are stored on flash memory for post-flight processing; "quick-look", near real time data for ozone, water, $N_2O$, $SF_6$, and $CH_4$ are also provided by a serial or Ethernet connection to the aircraft, for onboard use and telemetry to the ground.  Data are analyzed post-flight with home-built software, including GC peak integration and quantitation routines, primarily using Igor software, and three separate data files (GC, ozone and water, with different time intervals) are generated for archiving and dissemination.

**3 Field missions and modifications to UCATS**

**Table 1: Missions and configurations of UCATS.  A second 2B Model 205 ozone instrument was added for ATTREX-2 and 3.  The water vapor instrument was converted to the newer Port City model for ATom-2 and subsequent deployments.  For the DCOTSS mission (now scheduled to start in 2021), UCATS is being repackaged to include three GC channels to measure CFCs (CFC-11, CFC-12, and CFC-113) and H-1211; shorter-lived chlorine compounds ($CHCl_3$, $CCl_4$, and $C_2HCl_3$); and $N_2O$ and $SF_6$.**

| Mission | Year | Aircraft | GC configuration | Ozone | Water vapor |
|---|---|---|---|---|---|
| UAS Demo. | 2005 | Altair | CFCs; $N_2O$/$SF_6$ | 2B 205 | None |
| Western Fire | 2006 | Altair | $CH_4$/CO/$H_2$; $N_2O$/$SF_6$ | 2B 205 | Maycomm |
| START-08 | 2008 | GV | $CH_4$/CO/$H_2$; $N_2O$/$SF_6$ | 2B 205 | Maycomm |
| HIPPO | 2009-11 | GV | $CH_4$/CO/$H_2$; $N_2O$/$SF_6$ | 2B 205 | Maycomm |
| GloPac | 2010 | Global Hawk | $CH_4$/CO/$H_2$; $N_2O$/$SF_6$ | 2B 205 | Maycomm |
| ATTREX | 2011-15 | Global Hawk | $CH_4$/CO/$H_2$; $N_2O$/$SF_6$ | 2B 205 (2) | Maycomm |
| ATom | 2016-18 | DC-8 | $CH_4$/CO/$H_2$; $N_2O$/$SF_6$ | 2B 211, 205 | Port City |
| DCOTSS | 2021-22 | ER-2 | 3 channels, see caption | 2B 211 | Port City |

Aircraft missions that included UCATS are summarized in Table 1 and described in this section.  The first two projects were designed to show that high-quality measurements could be made on a UAS with autonomous instruments.  The NOAA/NASA UAS Demonstration Project using General Atomics' Altair (Predator B-ER)

UAS using General Atomics' Altair (Predator B-ER) UAS was conducted from Gray Butte, CA, during April-May and November, 2005. UCATS measured $N_2O$, $SF_6$, CFC-11, CFC-12, and halon-1211 and ozone. More than 60 hours of atmospheric composition data were acquired up to altitudes of 13 km, with the mission highlight an 18.4-hour flight over the eastern Pacific Ocean, successfully demonstrating that atmospheric composition and other environmental parameters can be measured with high precision and accuracy from a UAS [Fahey et al., 2006]. The NASA/USDA-FS/NOAA Western Fire mission was conducted in August and October, 2006, again using Altair [Hinkley et al., 2009]. The scientific focus was on remote mapping of wildfires from a UAS, with UCATS on board to measure atmospheric trace gases in fire plumes. At this point, the halocarbon GC channel was replaced with a channel to measure the combustion products CO and $CH_4$, along with $H_2$, and a TDL water vapor sensor was added to UCATS. Accomplishments of this project included 21- and 22-hour science flights and more than 65 hours of UCATS *in situ* measurements of trace gases and water vapor.

For START-08 [Pan et al., 2010] and HIPPO [Wofsy, 2011], both UCATS and the PAN and Trace Hydrohalocarbon ExpeRiment (PANTHER), a four-channel GC with ECD detection and a GC with mass spectrometry detection (GC/MS), were flown on the NSF/NCAR GV aircraft. These were integrated together with a NOAA whole air sampler (WAS) in collaboration with the University of Miami [Schauffler et al., 1999]. The larger GV payload was designed to probe long-lived greenhouse gases and tracers of atmospheric transport. In HIPPO, the GV flew repeated vertical profiles between 150 m above sea level and 14 km, largely over the Pacific Ocean, from northern Alaska and the Arctic Ocean to south of New Zealand near Antarctica, with five deployments from January 2009 to August 2011 covering different seasons. The first use of UCATS in the extremely humid tropics during HIPPO revealed several issues, which were resolved after the first two deployments. Initially, the GC columns adsorbed water, which changed their retention characteristics. To alleviate this, a Nafion dryer [Perma Pure, MD-050-72S-2] was added to remove most of the moisture from the GC air flow prior to the sample loops, with the exhaust from the pre-columns used as the dry counterflow gas. The Nafion dryer helped considerably, but retention times and sensitivity for the $N_2O/SF_6$ channel still showed changes after passing through very humid air; likely the Nafion dryer could not remove all the water vapor. As described in Moore et al. [2003], we use Porapak-Q columns to separate $N_2O$ and $SF_6$ from the large $O_2$ peak. This is followed by a short "post-column" of 5A molecular sieve, which partially retains the $N_2O$ but not $SF_6$, allowing the two peaks to be separated. The molecular sieves have a very high affinity to water vapor, and absorbed water changes the retention characteristics and peak height for $N_2O$. This problem was finally resolved by lengthening the 5A molecular sieve post-column from ~20 to 25 cm. This allowed the post-column to be operated at 190 °C instead of 115-120 °C, with $N_2O$ still completely eluting within 70 seconds after sample injection. Water does not accumulate in the post-column at the higher temperatures, and retention times and other aspects of the chromatography remained constant. The $N_2O$ peak also became much sharper and higher, improving the resolution of $SF_6$ and $N_2O$. This required changes to the electrometer circuit that processes the ECD signal, in order to achieve a faster response time and avoid saturation of the signal. These were completed in 2015, prior to the ATom mission. Plots illustrating data quality and intercomparisons are shown in Sect. 4.

Problems were also identified with the 2B ozone instrument during the START-08 and HIPPO missions related to changing humidity. UV ozone photometers are known to suffer from offsets when transitioning between wet and dry conditions [Wilson and Birks, 2006], because of water being retained in the scrubber and slowly released, differentially affecting reflectance from the walls of the cell with scrubbed air compared to the cell with ambient air. This was resolved for the ATom mission, as described later in this section.

From 2010 to 2014, UCATS was integrated into a compartment in the fuselage of the Global Hawk UAS for the GloPac and ATTREX missions [Jensen et al., 2013]. The Global Hawk generally operates in the stratosphere and upper troposphere (12-20 km), where air is very dry. However, these missions led to other changes in order to improve the data quality for ozone. The original 2B Model 205 could achieve a precision of $\pm2\%$ + 1 part per billion (ppb) with 10 second averaging at atmospheric pressure. Because Beer-Lambert absorption is really a measurement of number density (concentration) of the absorbing molecule, the precision varies inversely with pressure (1 ppb precision at 1000 hPa corresponds to a precision of 10 ppb at 100 hPa, a typical pressure in the UTLS). This is more than adequate for midlatitude and polar stratospheric missions such as GloPac, where ozone varies from a few hundred to a few thousand ppb. But for ATTREX, where the focus was the tropical tropopause layer (TTL; Fueglistaler et al., [2009]), ozone was typically less than 100

ppb, at pressures of 150-70 hPa. To partially address this issue in ATTREX-2 and 3 (2013-14), a second Model 205 sensor was added to UCATS. The original Model 205 remained completely enclosed and the new one was added to the front panel, with part of the instrument inside the sheet metal UCATS enclosure and the cell, lamp, and detectors on the surface, with a small insulated cover and warm airflow from UCATS passing through it. In general, the older 2B had better stability over a flight, possibly because of the more constant temperature environment inside UCATS. However, after a few hours of operation (always the case with Global Hawk flights, which could last for ~24 hours with the ATTREX payload), both instruments converged to stable and consistent readings. When both instruments were operating normally, data from the two instruments were merged and averaged to create a combined data set with a value reported every 5 seconds. By averaging the data to longer times (typically 10 seconds), the precision of the measurements could be improved. UCATS served as the primary ozone instrument during ATTREX-3, where weight and balance issues with the payload prevented the NOAA CSL instrument from being flown on the Global Hawk.

The ATom mission was similar to HIPPO, but with a much larger payload to map out and study atmospheric chemistry as well as long-lived gases over remote regions. For ATom, a new 2B Model 211 ozone photometer was added to UCATS and the partially external Model 205 removed. The Model 211 is similar in principle, but has a longer cell and path length (30 cm compared to 15 cm for the Model 205), improved electronics, and built-in flow meters to assure equal flows through each cell, with a stated precision that is the sum of 1% + 0.5 ppb over a 10-second average at 1000 hPa. As purchased, it used photolysis of $N_2O$ to produce NO as the ozone scrubber; this method is not affected by changes in humidity of the sampled air. However, at high altitudes, with fixed addition of $N_2O$ (or NO), the concentration of NO decreases with decreasing pressure in the instrument, and the rate of the chemical reaction ($NO + O_3 \rightarrow NO_2 + O_2$) that removes ozone decreases proportionally. Rather than trying to add more NO to compensate (carrying toxic gases like NO on an aircraft is problematic; even large amounts of an oxidizer such as $N_2O$ add to the complexity of getting a payload certified), we used $MnO_2$-coated screens as the scrubber and passed both the scrubbed and ambient airflows through the Nafion moisture exchangers provided with the instrument. Moisture exchangers have been shown to eliminate the artifacts associated with rapid changes in water vapor by keeping both cells at a constant humidity [Wilson and Birks, 2006]. They were not used for HIPPO and START-08, because the pressure of the gas flow being analyzed varied from ~100 to 1000 hPa while the cabin pressure is maintained near 900 hPa at high altitudes. With a pressure differential of over 700 hPa, the soft Nafion tubes could leak or collapse and block the flow. We solved this potential problem in ATom by placing the Nafion tubes in a small aluminum box (McMaster-Carr, 75895K series), sealed to the outside except for a small flow (50-200 sccm) of moist air (cabin air passed through a short piece of 12.7 mm dia. tube containing wet cotton) through the box and into the exhaust line from the ozone instrument. Thus, the pressure inside and outside the Nafion tubes stayed approximately equal. This setup adds moisture to dry air samples, and may actually remove some water from the very wettest samples (such as the tropical marine boundary layer, where the water content can exceed 3%), and generally keeps the humidity in the ozone instrument constant or at least the same in both the scrubbed and unscrubbed flows. This simple solution eliminated the effects of rapid changes in humidity, as demonstrated by comparisons with another ozone instrument (see Fig. 9 and 10 below).

The new ozone and water instruments were larger than the original models, and could not fit into the existing UCATS shell. A 7.5 cm extension was added to the top of UCATS, with the new water and ozone instruments and main cooling fans secured to the top plate. The original 2B Model 205 ozone instrument was left on the side to enable a comparison of results, and to provide a known and reproducible pressure measurement when needed. The total weight increase was about 5 kg, but this is negligible on an aircraft with the size and capacity of the DC-8. Starting in 2019, an additional repackaging and upgrade of UCATS has been carried out for flights on the ER-2 aircraft; this is described in Section 5.

## 4 Data intercomparisons and discussion

In this section, we present results in the stratosphere first, then in the troposphere.  To compare with UCATS, we used data from several other instruments.  On the GV, the Quantum Cascade Laser Spectrometer (QCLS; Santoni et al., 2014) measured long-lived trace gases, including $N_2O$ and $CH_4$, with high precision and 1-second time resolution, ideal for comparing time series and tracer-tracer correlation plots.  The PANTHER instrument (a 4-channel GC, with a separate GC/mass spectrometer) also measured the same molecules by GC as UCATS, using similar techniques.  Whole air samples were collected in glass flasks using Programmable Flask Package units (PFPs), which could be filled on demand or in a preset sequence, with 24 samples typically collected per flight.  Samples were later analyzed at the NOAA Global Monitoring Laboratory for a large set of trace gases [Sweeney et al., 2015].  All these instruments were also on the DC-8 aircraft for the ATom mission.  Data from the Airborne Chromatograph for Atmospheric Trace Species (ACATS; Elkins et al., 1996), a predecessor of both PANTHER and UCATS, are also used from the 1997 Photochemistry of Ozone Loss in the Arctic Region in Summer (POLARIS) mission for reference.

The NOAA Chemical Sciences Laboratory (CSL) "Classic" ozone instrument [Proffitt and McLaughlin, 1983] has a long history of measurements on high altitude aircraft, and flew on the GV during HIPPO.  This was replaced with a new lighter version, "NOAA-2" [Gao et al., 2012] for Global Hawk missions.  A different group from NOAA CSL flew a chemiluminescence ("CL") instrument on the DC-8 in ATom for measurements of ozone [Bourgeois et al., 2020], NO, $NO_2$, and total reactive nitrogen ($NO_y$).  Ozone data from concurrent GML sonde launches [Komhyr et al., 1995] and the NCAR chemiluminescence instrument [Ridley et al., 1992], on the GV during the Convective Transport of Active Species in the Tropics (CONTRAST) mission, were also used for ATTREX data comparisons and analysis.

The diode laser hygrometer (DLH; Diskin et al., 2002; Podolske et al., 2003), an open-path near-infrared absorption instrument, whose optical path is defined by a transceiver in the fuselage and a retroreflector mounted below one of the wings, was used to measure water vapor on the Global Hawk and DC-8.  During ATTREX, the NOAA CSL TDL hygrometer (also from Port City Instruments) measured water vapor as well.  The Meteorological Measurement System (MMS; Scott et al., 1990) was used on the Global Hawk and DC-8 missions for position and meteorological variables.

### 4.1 Gas Chromatographs

Global Hawk flights during the GloPac mission covered a wide range of air masses in the stratosphere, and provided an opportunity to demonstrate the capabilities of UCATS in the environment for which it was designed.  Figure 2 shows a scatter plot of $SF_6$ vs. $N_2O$ mole fractions for the flight of April 23, 2010, from Edwards AFB, CA to the western Arctic Ocean and back (~35-85 °N, 120-165 °W) at altitudes from 16 to 20 km, with two profiles down to 13 km and back.  $N_2O$ and $SF_6$ are long-lived greenhouse gases emitted at Earth's surface, and generally decline with altitude in the stratosphere [e.g., Plumb and Ko, 1992].  For $N_2O$ this is primarily because of photochemical loss in the stratosphere, and for $SF_6$ it is because older air entered the stratosphere at earlier times, when tropospheric $SF_6$ mixing ratios were lower [Hall et al., 2011].  As a result, $N_2O$ and $SF_6$ are correlated in the stratosphere, with older air and air from higher altitudes having the lowest mixing ratios for both gases.  This correlation can be seen in Fig. 2, where $N_2O$ declines strongly from its tropospheric value (~320 ppb in 2010) as $SF_6$ (tropospheric value ~7 ppt in 2010) approaches 5.5 ppt.  Data from ACATS-IV taken on the ER-2 aircraft almost 13 years earlier in the Arctic during the 1997 POLARIS mission are shown with the GloPac data for reference.  POLARIS $N_2O$ and $SF_6$ mixing ratios were adjusted upward for the tropospheric growth over the 13 years between missions ($N_2O$ increased from 312.5 to 322.9 ppb, and $SF_6$ from 3.9 to 7.0 ppt) by adding the difference in $SF_6$ tropospheric values to the POLARIS data, and multiplying the ratio of tropospheric values to the POLARIS data for $N_2O$ and other tracers that are photochemically destroyed in the stratosphere.  This is mainly to bring the two data sets onto the same scale for easy visualization, though the similarity does reflect the relatively stable nature of stratospheric circulation and photochemistry.  Average tropospheric values of these long-lived greenhouse gases were taken from the NOAA GML network (www.esrl.noaa.gov/gmd/ccgg/trends/global.html).  The measurement precision for UCATS is about ±0.05 ppt $SF_6$ (1 σ) and ±1.5 ppb N2O, similar to ACATS, but with a data rate of every 70 seconds instead of 360 seconds.  The slightly more gradual slope for POLARIS data is

due to the fact that SF$_6$ was increasing more slowly in the 1990's than in the years just before GloPac [Hall et
al., 2011], and perhaps other more subtle differences in the trends.  Similar plots of GloPac and (adjusted)
POLARIS CH$_4$ vs N$_2$O and H$_2$ vs CH$_4$ data from the same flights (Fig. 3) show close correspondence between
the two campaigns and tight, nearly linear correlations, as expected from the fact that these are all long-lived
gases in the stratosphere.  Overall, UCATS precision for CH$_4$ and H$_2$ was ±7-8 ppb (0.5%) and ±5 ppb (1%)
respectively, equal to or slightly better than that of ACATS-IV, and with a slightly faster data rate.

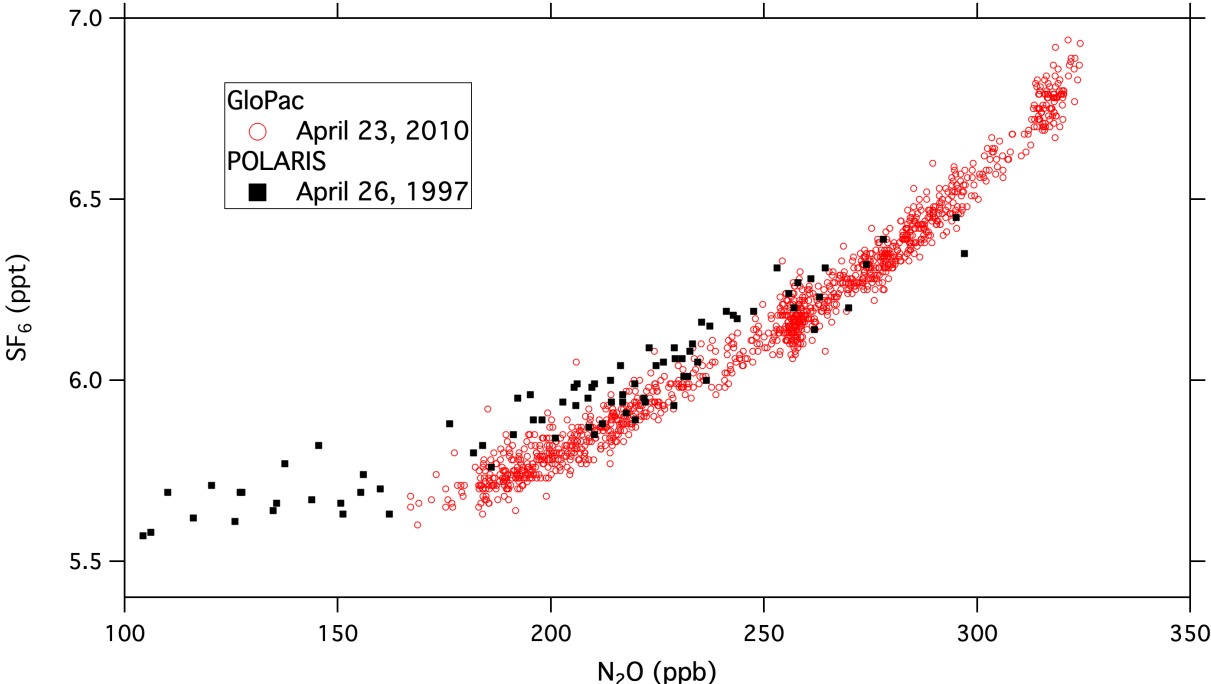

**Figure 2: UCATS SF$_6$ plotted vs. N$_2$O from the GloPac mission (red circles), with similar data from ACATS (black**
**squares) during the POLARIS mission 13 years earlier.  The POLARIS data have been adjusted for the**
**tropospheric increases in both gases between 1997 and 2010 (see text).**

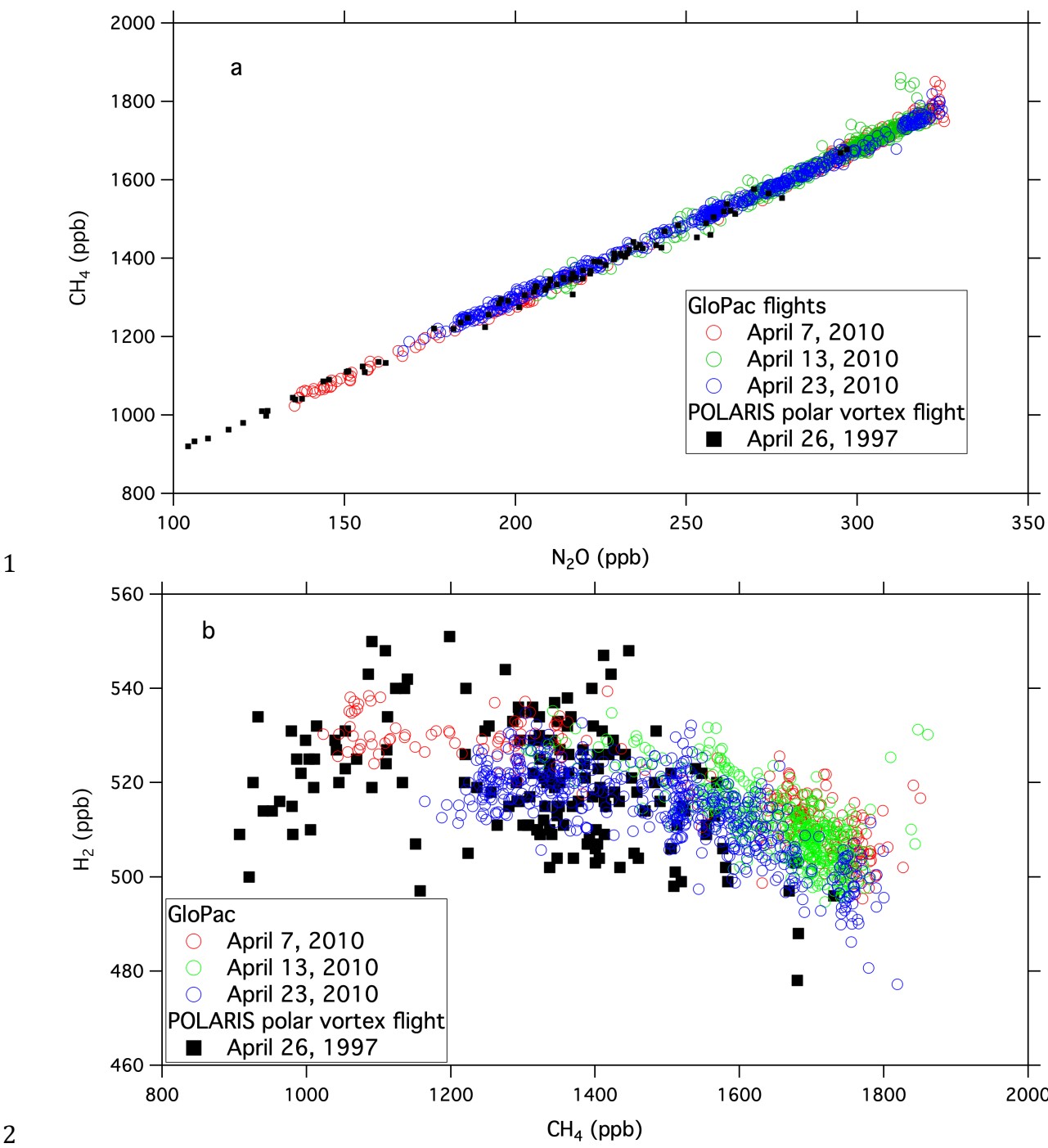

Figure 3: Similar correlation plots to Fig. 2, showing CH$_4$ vs. N$_2$O (a) and H$_2$ vs. CH$_4$ (b). Molecular hydrogen increases slightly in the stratosphere from CH$_4$ photooxidation, leading to their anticorrelation.

The START-08 and HIPPO missions were the first tropospheric campaigns for UCATS. On January 12, 2009, during the first HIPPO deployment, the GV sampled air in both the troposphere and stratosphere as it traveled from Anchorage, AK, north to near 80 °N over the Arctic Ocean, and back. The precision for UCATS N$_2$O during most of the flight was near ±1 ppb in both the troposphere and stratosphere, calculated from flight segments with near-constant N$_2$O near the start of the flight, and comparisons with QCLS and PFP samples throughout the flight (Fig. 4a). A more quantitative comparison can be made by plotting UCATS and

PFP data for the entire HIPPO 1 deployment against the higher time resolution QCLS data (Fig. 4b). Each UCATS GC measurement is a roughly 2-second average of the atmospheric composition along the flight path a few seconds before the air sample is injected, and is plotted here against the corresponding 10-second average of QCLS data. Each PFP flask takes between 30 seconds and a few minutes to fill, depending on altitude, and a comparison with QCLS data is enabled by averaging the QCLS data over the sampling interval associated with each flask sample. QCLS data have been corrected here for the approximately 1 ppb offset with respect to the PFPs reported by Santoni et al. (2014) during HIPPO in 2009-2011. The UCATS vs. QCLS correlation allows an upper limit estimate of UCATS precision, assuming all the error is associated with the UCATS measurements, none from QCLS, and that effects related to atmospheric variability arising from timing mismatches are negligible. The resulting standard deviation (1-$\sigma$ precision) is $\pm$1-2 ppb over the entire month of HIPPO flights, from the high Arctic through the tropics to the Southern Ocean and back. The slope of the fit is 0.91 $\pm$ 0.004; this difference has not been resolved. We note that the slope for the PFP data is 0.93 $\pm$ 0.02, though this is partially driven by the smaller slope for tropospheric (high $N_2O$) data, as opposed to for UCATS, where the slope is also smaller in the troposphere but clearly reflects differences in the stratosphere (low $N_2O$), where the dynamic range of $N_2O$ is large. The UCATS and PFP results agree closely over the more limited range of PFP $N_2O$ data, but because the measurements were not simultaneous, a quantitative comparison is not possible.

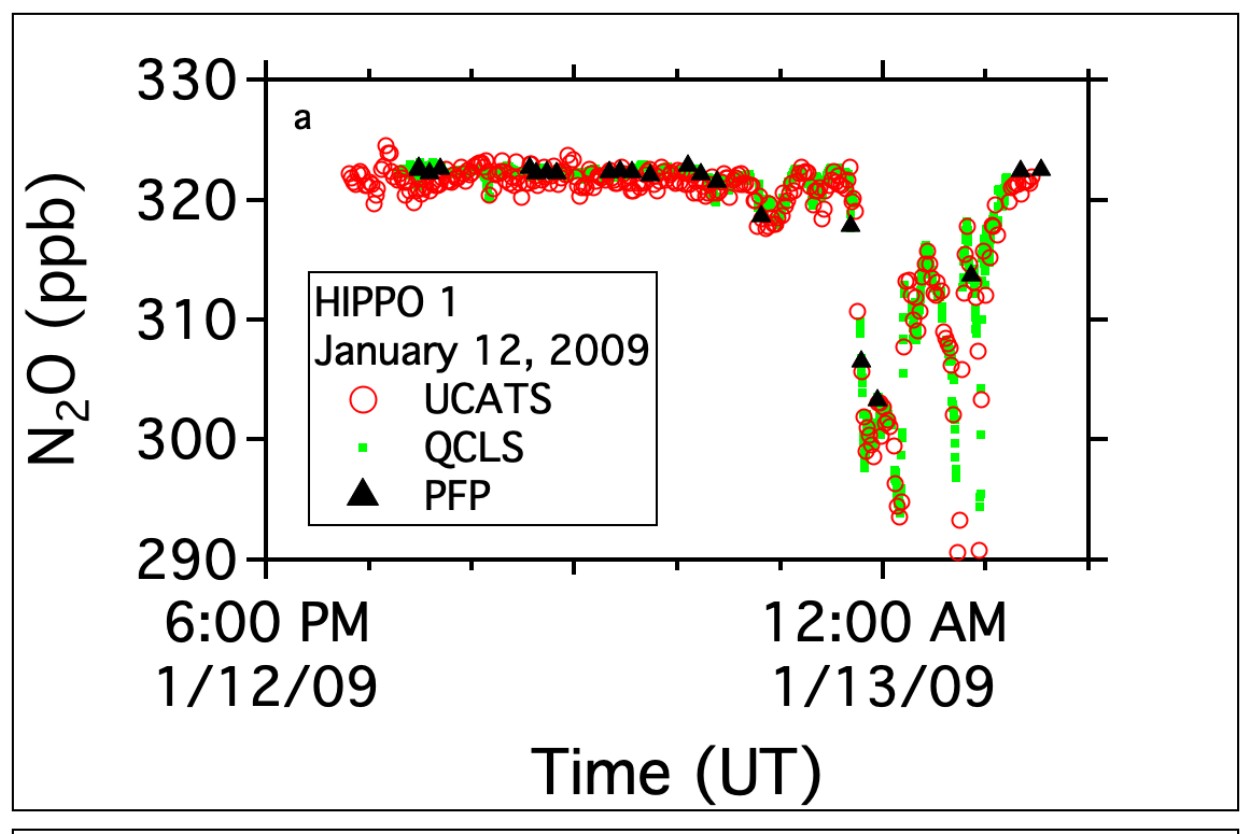

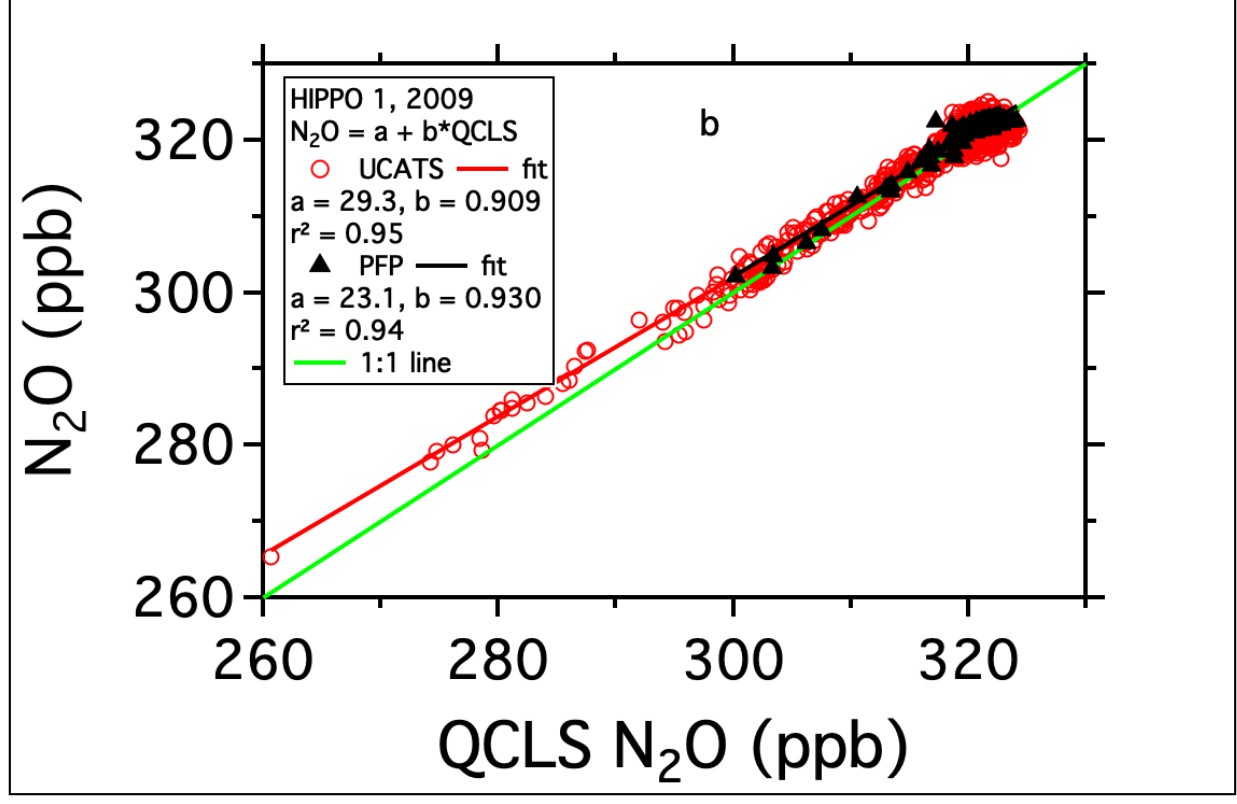

**Figure 4:** $N_2O$ data from the HIPPO mission on the GV aircraft.  Panel (a) shows the time series from a flight north from Anchorage, AK over the Arctic Ocean and back.  The aircraft flew several profiles from 14 km to near the surface during the flight.  Panel (b) shows UCATS and PFP data plotted against QCLS data over the entire deployment.

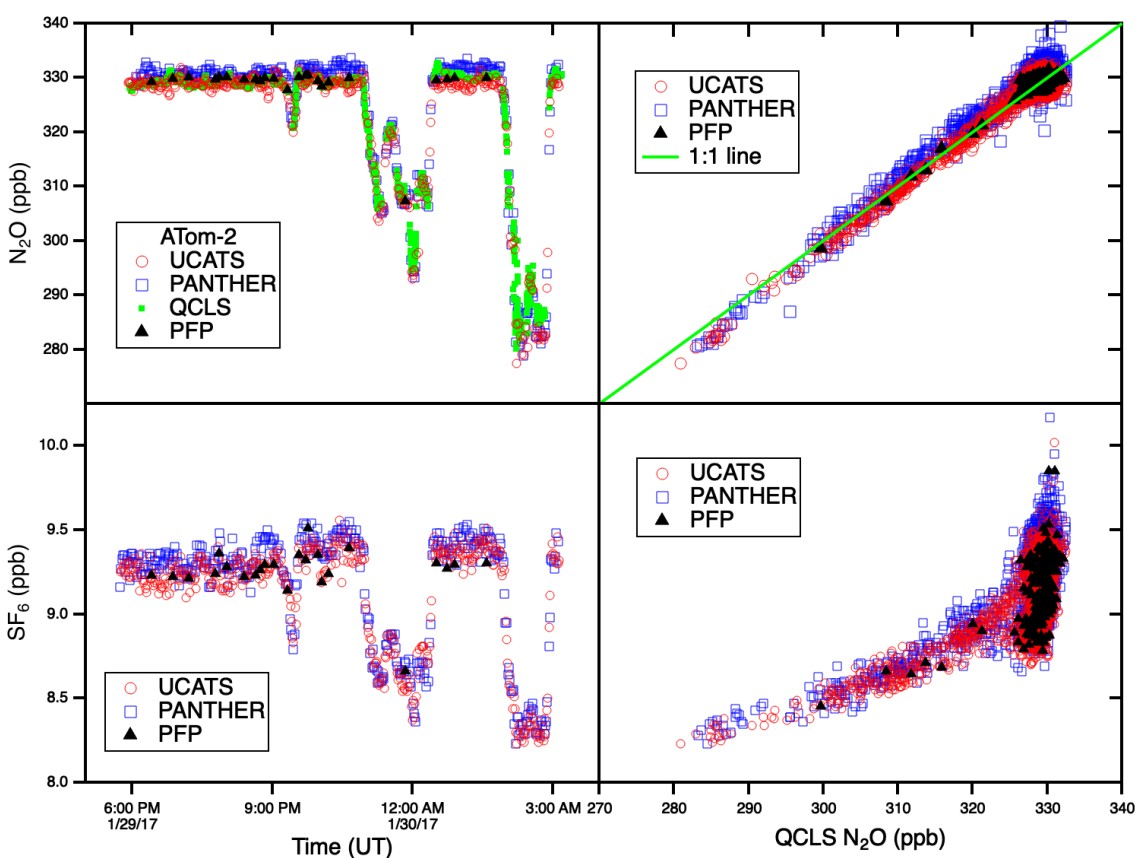

**Figure 5:** $N_2O$ and $SF_6$ time series for one flight (left), and scatter plots for all the NOAA instruments against QCLS $N_2O$ data from the entire ATom-2 deployment (right).  The flights, from January 26 to February 21, 2017, spanned the Pacific, Atlantic and Southern Oceans, and the Arctic.  With improvements to the $N_2O/SF_6$ channel, UCATS was able to achieve similar precision to the HIPPO-1 data shown in Fig. 4, but without the instability driven by accumulated water in the GC columns.  $N_2O$ and $SF_6$ do not have a simple relationship in the troposphere, so the panel on the lower right does not yield a single curve as in Fig. 2.

In ATom, with the GC system optimized for all conditions, UCATS produced precise and accurate data in the troposphere with both short and long-term stability and without degradation in the humid tropics.  This is demonstrated by $N_2O$ and $SF_6$ time series for the DC-8 flight of January 29, 2017, from Palmdale, CA to northern Alaska, and back to Anchorage (~35-70 °N, 120-155 °W), and scatter plots for the entire ATom-2 deployment (Fig. 5).  Data from UCATS, PANTHER, QCLS ($N_2O$ only), and PFPs show excellent agreement for the time series (mean differences typically ±1-2 ppb $N_2O$ and ±0.05 ppt $SF_6$).  ATom QCLS $N_2O$ data show a similar offset relative to PFPs (taken as the reference instrument on board) as observed during HIPPO, attributed to the QCLS calibration procedure.  As in Gonzalez et al. [2021], QCLS $N_2O$ data are corrected by subtracting the offset with respect to the PFP data (~1.2 ppb).  The 1-σ precision of UCATS and PANTHER was ~±1 ppb $N_2O$ and ±0.05 ppt $SF_6$ from both the time series and the scatter plots (again assuming all the variability in the comparison with QCLS is associated with the GC data).  As described above, $SF_6$ and $N_2O$ are well correlated in the stratosphere, and the precision of $SF_6$ from the lower right panel can be estimated for stratospheric data (lower values of both gases).  In the troposphere, the plot reflects the strong latitudinal

gradient in $SF_6$, with lower $SF_6$ in the southern hemisphere and higher $SF_6$ in the northern hemisphere.  This
leads to a much steeper apparent slope, since $N_2O$ also has a latitudinal gradient, but weaker, also with lower
values in the southern hemisphere.  Transitions between the troposphere and the stratosphere lead to mixing
lines between the two branches (from ~310-330 ppb $N_2O$).  The only disagreement for $N_2O$ is at low values,
where PANTHER and UCATS both measure about 3 ppb lower than the QCLS instrument, a deviation in the
opposite direction compared to HIPPO.  The tropical flight of February 3, 2017 (Fig. A3) illustrates the
precision of $N_2O$ and $SF_6$ where air masses sampled along the flight track varied slowly (because of its altitude
range, the DC-8 is always in the troposphere at these latitudes).   $H_2$ measurements also showed good
agreement between UCATS, PANTHER, and PFPs (Fig. 6), with nearby data points from the different
instruments typically differing by about ±5 ppb (1%) over the entire range of observed values.  Values for
precision and agreement of measurements from ATom and other missions are summarized in Table 2.

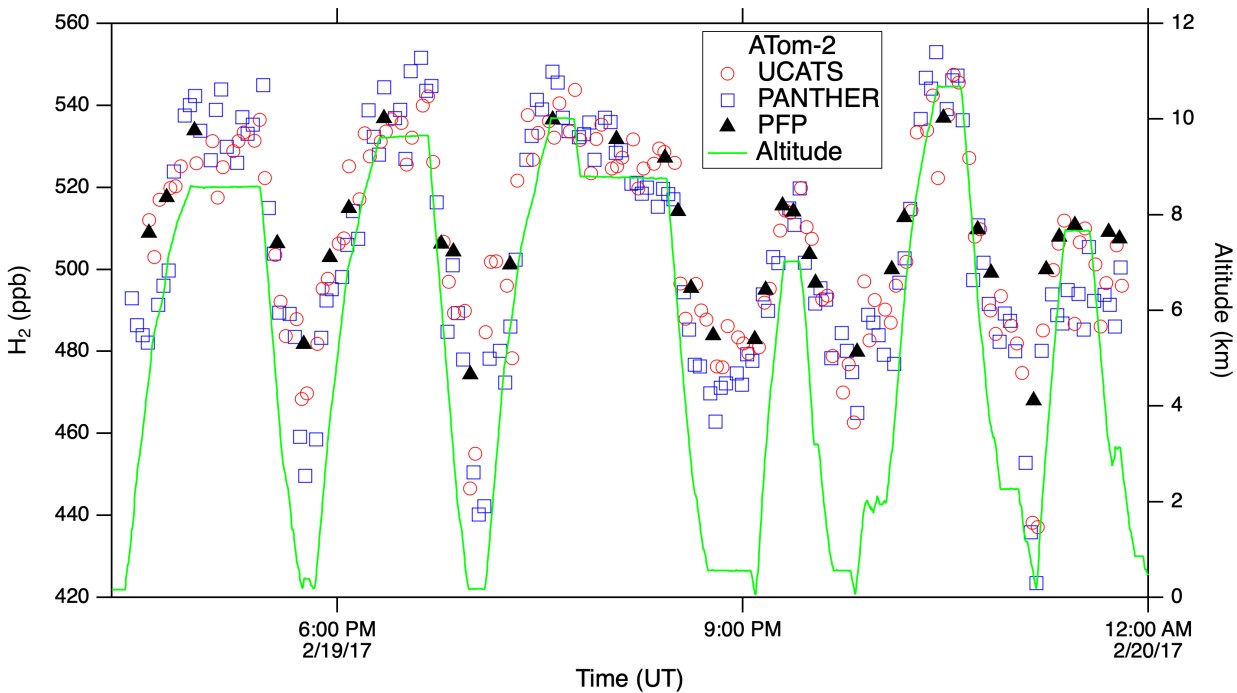

**Figure 6: $H_2$ from UCATS, PANTHER, and PFPs showed close agreement in the troposphere and lower**
**stratosphere during ATom.  Because of the dominant soil sink for $H_2$, mixing ratios are lower near the surface, as**
**seen on this flight from Thule, Greenland to Anchorage, AK, over the Canadian Arctic, the Arctic Ocean, and**
**Alaska.**
**Table 2: Precision of UCATS measurements for selected missions, level of agreement with other instruments on**
**the same platform, and comparison of lower tropospheric values of long-lived gases with the NOAA surface**
**network.  The NOAA surface network of flask collection sites allows the airborne measurements to be tied to a**
**global system with calibration scales for all the gases measured by GC.  The NOAA airborne instruments use**
**standards from the same laboratory as the surface network, as well as the same scales.  CO is not included**
**because it was usually measured by other instruments, and has an artifact in the stratosphere at high ozone**
**levels.  In the troposphere the precision for CO was typically 5-10 ppb.  The main sink of $H_2$ is at the surface and it**
**has the most variability there, so we do not compare it with the surface network.  Not all deployments and**
**measurements achieved the same level of precision as shown here.  This table applies to HIPPO 1 for long-lived**
**gases; GloPac and ATTREX-2 and 3 for $N_2O$ and $SF_6$; and ATom-2, 3, and 4 for all species measured by UCATS.**
**[a]For these measurements, UCATS agreed to 1-2 ppb with PFP whole air samples; we consider these to be the most**
**precise and accurate measurements of $N_2O$.  Agreement with QCLS in the troposphere was also 1-2 ppb, but**
**diverged for $N_2O$ < 300 ppb, with differences up to 4 ppb.  The goal of HIPPO was to quantify long-lived tracers**
**and other greenhouse gases in the troposphere, with less emphasis on the stratosphere, and the calibration gases**
**carried on board by QCLS spanned the range of about 290-330 ppb.  Because UCATS is calibrated with a range of**
standards from 160-322 ppb in HIPPO, we are confident in our measurements throughout the stratosphere,
though these calibrations are performed on the ground and only one standard (with mixing ratios close to the
background troposphere) is used in flight.
[b]Precision for water vapor is best expressed as a percentage in the troposphere (high water vapor) and an
absolute number in the stratosphere (low water). The precision here is given for the lowest water vapor values
in the stratosphere, 1 ppm for the original MayComm instrument and 0.1 ppm for the newer Port City instrument.
The precision in the troposphere was always 5% or better. Calibration and comparison of the Port City
instrument are still ongoing.
[c]The precision and agreement for ozone are best expressed in ppb at low mixing ratios, and as a percentage at
high mixing ratios. For example, in ATTREX and GloPac, where ozone ranged from less than 20 ppb to over 2000
ppb, the average agreement with the CSL instrument was better than 1 ppb for ozone < 200 ppb, with a precision
better than 10 ppb, and for ozone > 500 ppb, the average agreement was better than 1% with a precision of 2-3%.
Values for ATom are for tropospheric data ($O_3$<100 ppb), since the focus of ATom was on tropospheric chemistry.

| Mission | Year | Molecule | Precision | Agreement with onboard instruments | Agreement with surface network |
|---|---|---|---|---|---|
| HIPPO | 2009-11 | N2O | 1.5 ppb | 2 ppb[a] | 1 ppb |
| | | SF6 | 0.05 ppt | 0.05 ppt | 0.03 ppt |
| | | CH4 | 15 ppb | 10 ppb | 15 ppb |
| | | H2 | 5 ppb | 5 ppb | — |
| | | O3 | 9 ppb | 3 ppb | — |
| | | H2O | 1 ppm[b] | 5% | — |
| GloPac/ | 2010 | N2O | 1 ppb | — | — |
| ATTREX | 2011-15 | SF6 | 0.04 ppt | — | — |
| | | CH4 | 7.5 ppb | <5 ppb | — |
| | | H2 | 5 ppb | — | — |
| | | O3 | 5-10 ppb[c] | 1% or 1 ppb[c] | — |
| | | H2O | 1 ppm | 5% | — |
| ATom | 2016-18 | N2O | 1 ppb | 1.5 ppb | 0.8 ppb |
| | | SF6 | 0.05 ppt | <0.05 ppt | 0.04 ppt |
| | | CH4 | 15 ppb | <5 ppb | 10 ppb |
| | | H2 | 5 ppb | 5 ppb | — |
| | | O3 | 2-3 ppb | <1 ppb | — |
| | | H2O | 0.1 ppm | in process | — |

**4.2 Ozone - stratosphere**
This section is primarily focused on the ATTREX mission, which was designed to probe the chemical
composition of air over the tropical Pacific and transport into the stratosphere, but applies to all UCATS
stratospheric data. Because ozone mixing ratios peak in the stratosphere, the main requirements for a
stratospheric ozone measurement are accuracy and stability, with sensitivity to low values usually less
critical. In GloPac, the 2B Model 205 in UCATS agreed within 1% with NOAA Classic over the large observed
range of ozone mixing ratios (Fig. A4). However, as discussed in section 3, requirements are different in the
TTL, where ozone mixing ratios are very low, often less than 30 ppb. Both measurement accuracy and
precision are essential at these low values, and even errors of a few ppb in ozone (or small measurement
biases in water vapor and other trace gases) can lead to different interpretations of the underlying
atmospheric processes. The accuracy of the Model 205 ozone instrument can be calculated similarly to
Proffitt and McLaughlin (1983), where the most important uncertainties are the absorption cross-section of
ozone, the accuracy of cell temperature and pressure measurements, the absorption path length, and any
nonlinearities in detector response.  These add up to a few percent, but the initial calibration of the 2B
instruments against a reference standard (by 2B) should correct for any slight inaccuracies.  In all our
calibration checks, the slope was within 1% of unity and the offset less than 2 ppb (usually <1 ppb), at
ambient pressure (~840 mbar in Boulder, CO, and 920 mbar in California) and room temperature.
As described in Sect. 3, two Model 205 sensors were flown in UCATS during ATTREX-2 and 3, with data from
the two instruments merged into a single data set with higher time resolution than the original instrument.  A
comparison of UCATS and NOAA-2 ozone data from ATTREX-2 (Fig. 7) shows that the slope is close to unity
with a crossover point near 500 ppb.  At low ozone (20-30 ppb), the UCATS data are on average lower by 3-4
ppb.  Since the absorption cross-sections are the same for both instruments and cell length is fixed (and
measured to better than 1% accuracy), the principal known sources of error are inaccuracies in measured cell
temperature and pressure.  The pressure sensor in the older 2B instrument was carefully calibrated over a
range of pressures for many years (2010-2016), and was stable throughout that time.  A small correction was
made to account for the pressure drop from the cell to its outlet (where pressure is measured).  This
introduced about a 1% increase in ozone at the highest altitudes but was negligible at lower altitudes.
Temperature is measured on the cell body rather than in the airflow, but air temperature should have time to
equilibrate inside UCATS before reaching the ozone instruments. (Flow to each 2B is ~10% of that for NOAA-
2, which has been shown to measure temperature accurately after warming ambient air as it flows to the cell
[Gao et al., 2012].)  The offset between NOAA-2 and 2B data bears further examination.  UV ozone
photometers have been shown to produce offsets when transitioning from wet to dry conditions (Wilson and
Birks, 2006), and that is certainly the case for the Model 205, as discussed in the following section.  However,
except on initial ascent, air sampled in ATTREX was always extremely dry, and any artifact should become
negligible within 1 hour.  Similar agreement between NOAA-2 and the original 2B instrument was obtained
on ATTREX-1 and GloPac.  Laboratory tests for measurement artifacts of the 2B under various conditions
produced mostly negligible offsets, and always less than 5 ppb.

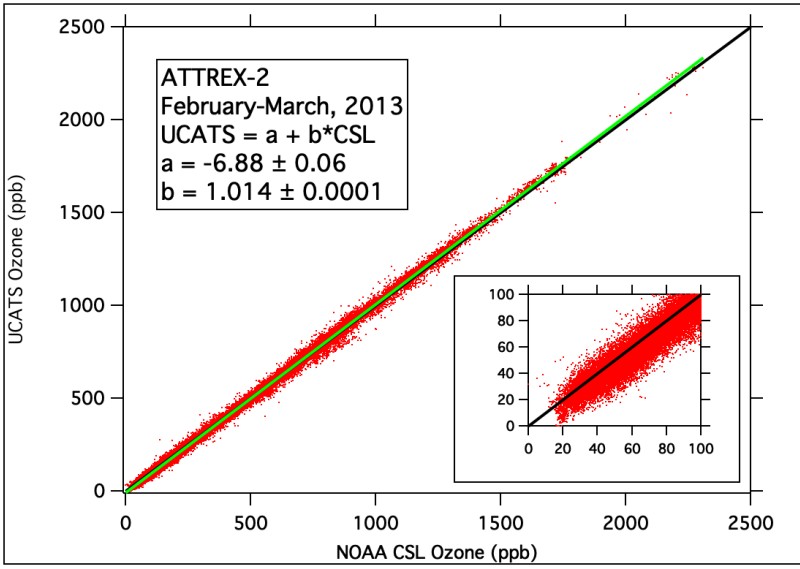

**Figure 7: Merged UCATS ozone data from the two 2B Model 205 instruments plotted against coincident data from**
**the NOAA Chemical Sciences Laboratory NOAA-2 ozone instrument for all six research flights during ATTREX-2 in**
**2013.  The green line is the least squares fit to the data (parameters shown in the legend) and the 1:1 line is**
**shown in black.  The inset shows ozone values <100 ppb.**
During ATTREX-3, payload weight and balance issues prevented the NOAA-2 instrument from being flown on
the Global Hawk.  Coincident balloon-borne electrochemical concentration cell (ECC) ozonesonde launches
from Guam provide a comparison for these flights.  Data from the last 2 hours of the February 16-17, 2014

Global Hawk flight (Fig. 8) most closely overlapped one of the balloon profiles in space and time (within 100 km and 1-2 hours). The agreement between the ECC and the 2B instruments in the troposphere (<16 km, where the balloon and aircraft were in closest proximity) is quite good and shows no significant bias in the UCATS data. A further check on UCATS ozone is shown in Fig. A5 with ozone data from the GV aircraft (operating during the concurrent CONTRAST mission; Pan et al., 2017), the Global Hawk, and the Guam ozonesonde launch, which was timed to overlap with the return of the Global Hawk on February 13. In summary, based on laboratory calibrations, tests, and in-flight comparisons, we assign a systematic uncertainty of less than 5 ppb to our Model 205 ozone data in the TTL and lower stratosphere. The precision in the TTL ranged from ±5 to 10 ppb but can be improved by temporal averaging. The low values of ozone in the TTL demonstrate the importance of precise and sensitive ozone measurements in this region and the need to minimize or eliminate any systematic errors.

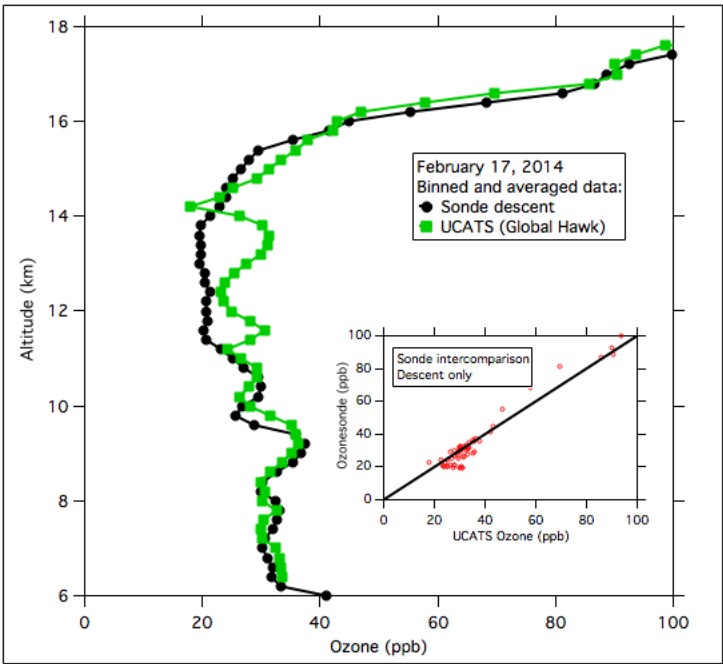

**Figure 8: UCATS and ozonesonde data from February 17, 2014 near Andersen AFB, Guam. UCATS data (solid green squares) from the last part of the flight are binned and averaged by altitude; the solid black circles are the binned and averaged sonde descent data from near when the Global Hawk landed. The inset shows a scatter plot of binned UCATS and sonde data at coincident altitudes; the 1:1 line is shown in black.**

### 4.3 Ozone - troposphere

As described above, the Model 205 in UCATS disagreed with the NOAA "Classic" instrument during HIPPO following transitions between wet and dry air. Most flights had only minor artifacts, but the issue was most pronounced in the tropics, with an example shown in Fig. 9 (top). At low altitudes there was generally good agreement (mean difference = 0.4 ppb, standard deviation = ±4.2 ppb for HIPPO 4), but as the GV aircraft climbed out of the very wet lower troposphere to higher altitudes, or descended back into the lower troposphere, changes in water retained in the scrubber likely affected the reflected light along the sides of the cell, causing anomalous ozone readings. Even though flows were greater than 1 liter/min., the instrument took 15 minutes or more to recover. In ATom, with the newer Model 211 instrument and moisture exchangers for both scrubbed and unscrubbed air, the agreement was much closer over a similar flight track, and there were no anomalous data segments as the DC-8 ascended and descended (Fig. 9, bottom). The one discrepancy is in the tropical marine boundary layer (MBL), where UCATS was typically a few ppb higher than the chemiluminescence instrument. This disagreement is outside the combined uncertainties of the two instruments and is not currently understood; UCATS showed no differences when calibrated with wet or dry air in the laboratory, and the effect of water vapor on the chemiluminescence instrument has recently been

re-checked. There were no offsets in the high latitude MBL (agreement within 1 ppb; see Fig. A7), so it is
presumably related to the high humidity or something else present in the tropical MBL.

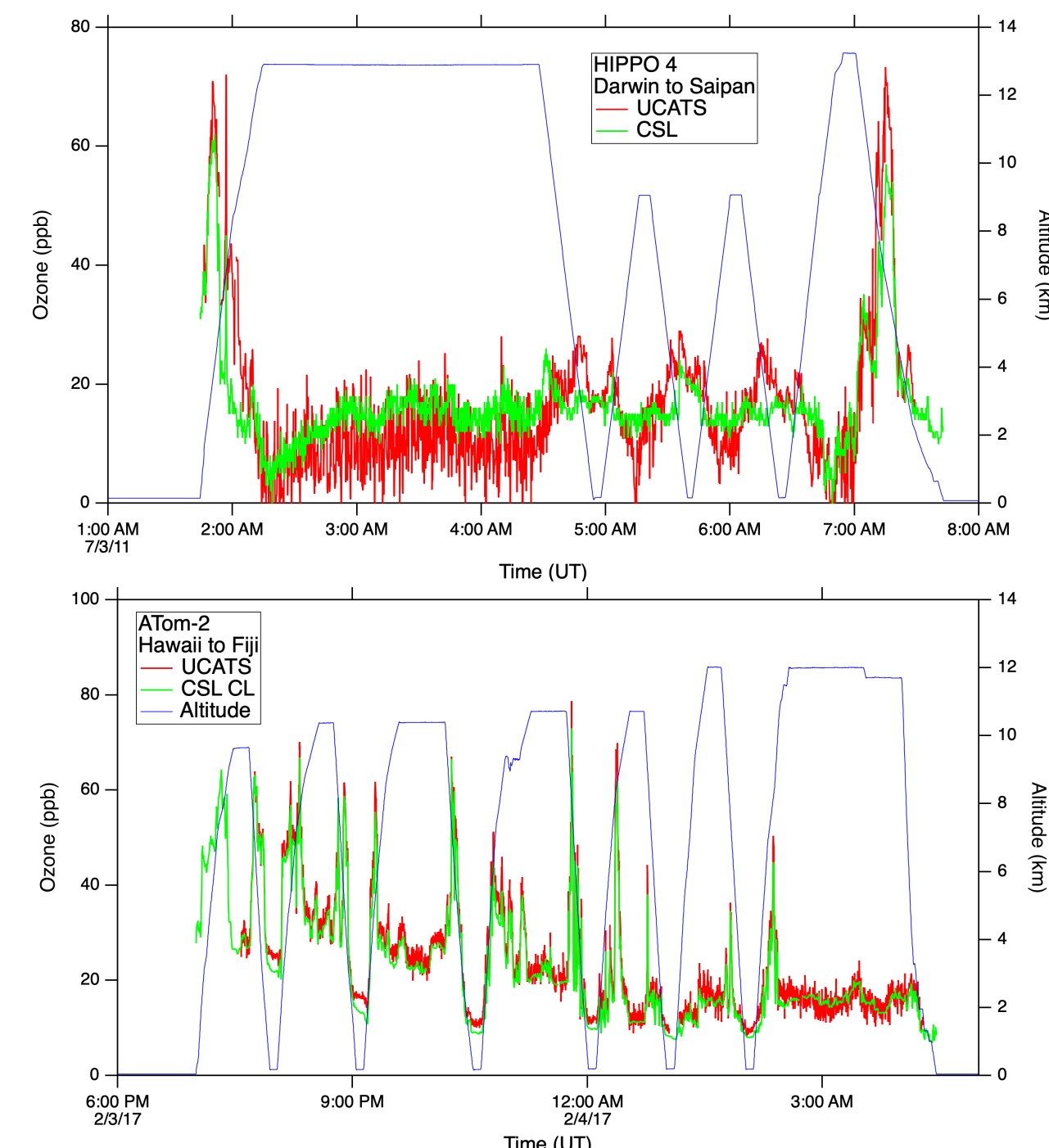

**Figure 9: Ozone time series from HIPPO (top) and ATom (bottom). The upper plot shows an example of the**
**discrepancies between the 2B Model 205 and Classic Ozone instruments observed in the tropics during HIPPO;**
**extratropical flights were in closer agreement. The lower plot shows ATom data (over a similar range of**
**latitudes) from the NOAA chemiluminescence instrument and the UCATS Model 211 ozone instrument, modified**
**with moisture exchangers to ensure that air to both cells remained at reasonably high humidity.**

Scatter plots of UCATS ozone data against the corresponding instrument from NOAA CSL (Fig. 10) showed reasonable overall agreement in HIPPO, with the slope within 1% of unity. But in HIPPO, there are many UCATS data points in the troposphere with significantly higher ozone than measured by the Classic Ozone instrument (e.g., near 50 ppb). The improvement between HIPPO (top) and ATom (bottom) is dramatic. This is partly due to the longer optical path length in the Model 211, as well as other instrumental improvements, but the addition of the Nafion moisture exchangers makes a substantial difference as the aircraft transitions between wet and dry air masses. It should be noted that the NOAA CSL instruments being compared to here have completely different designs – Classic Ozone is a UV photometer like the 2B, while in ATom, ozone was detected by chemiluminescence, though it is fundamentally calibrated using an optical measurement. Both of these instruments have a precision in the troposphere of about ±0.5 ppb. The larger deviations occasionally observed in the ATom data are mostly due to timing mismatches during flight segments with sharp gradients in ozone, along with occasional outliers from all instruments (see Fig. A6). The older Model 205 was also flown during ATom as a backup and for comparison with the Model 211; the Model 205 showed some of the same deviations between wet and dry air as in HIPPO, while the Model 211 with Nafion moisture exchangers tracked the CSD instrument closely. For many applications, such as climatologies, chemical modeling, and transport studies in the troposphere, the precision of the Model 211 (±0.5 ppb at sea level) as flown during ATom is more than adequate, given the good overall agreement with the chemiluminescence instrument. In the stratospheric parts of the ATom flights, the Model 211 instrument had precision of about ±1% and agreement within 2% (not shown).

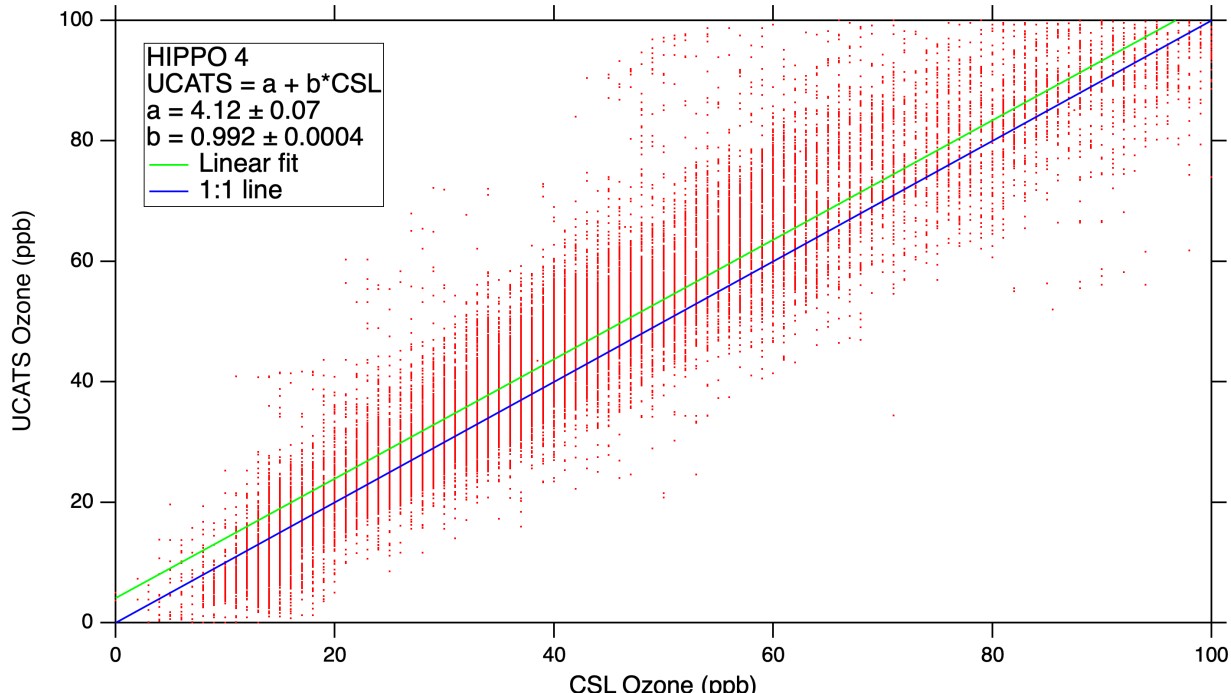

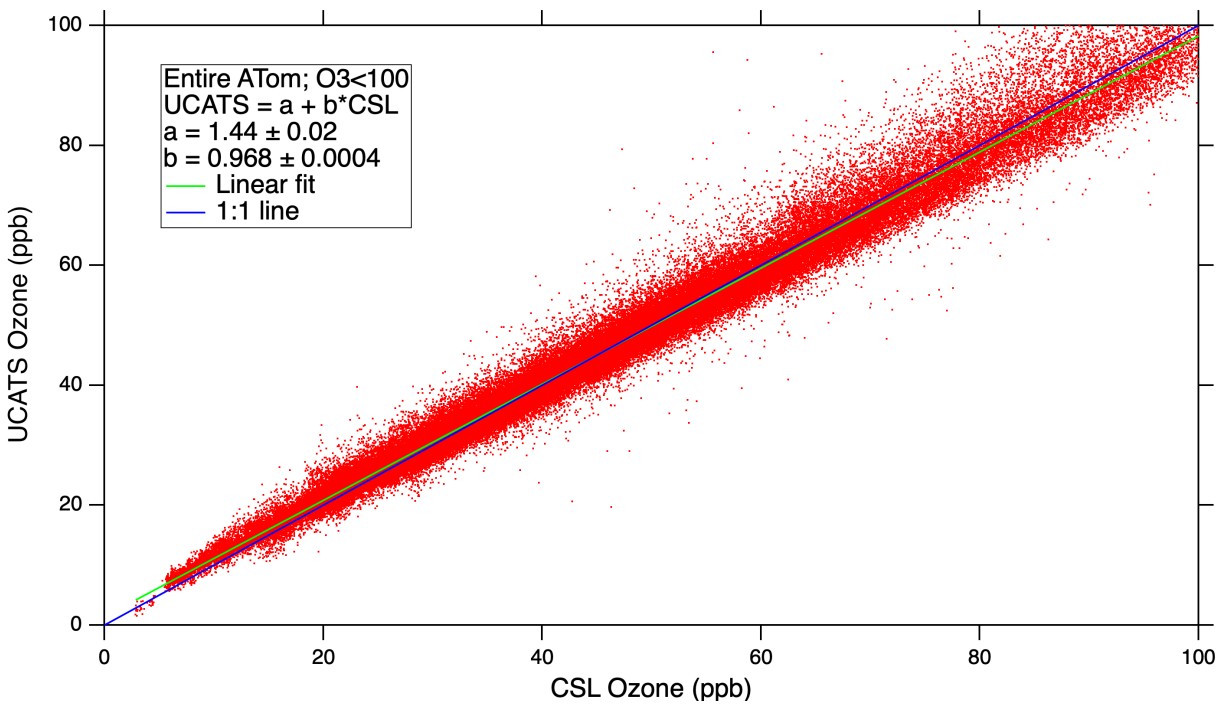

Entire ATom; O3<100
UCATS = a + b*CSL
a = 1.44 ± 0.02
b = 0.968 ± 0.0004
— Linear fit
— 1:1 line

**Figure 10: Scatter plots of UCATS ozone (2B Model 205) vs. CSL "Classic" ozone (top) and UCATS ozone (2B Model 211) vs. CSL chemiluminescence ozone (bottom) from tropospheric observations.**

## 4.4 Water vapor

The original MayComm TDL instrument was used in UCATS from 2006 to the first ATom deployment in July-August 2016.  Its uncertainty was the sum of 5% + 1 ppm, based on laboratory calibrations with gravimetrically-prepared standards and frost point hygrometers.  The ±1 ppm precision limit made stratospheric measurements above ~16 km (where water vapor is typically 2-8 ppm) somewhat qualitative compared to the troposphere and lower stratosphere ($H_2O$ > 10 ppm).  In addition, there was a temperature effect on the electronics of the long-path channel (for low water vapor) such that the sensitivity dropped with increasing temperature inside UCATS, up to 30% in extreme cases.  This was addressed by adding a Peltier cooling circuit to the TDL electronics box during HIPPO, which kept it at 25 °C (except on occasional cold or very warm takeoffs), and also by calibrating and correcting for the temperature effect.

For ATom (starting with deployment 2, January-February 2017), we integrated a new, larger TDL instrument from Port City Instruments, the successor to MayComm.  The longer wavelength (2.574 μm vs. 1.37 μm) utilizes stronger absorption lines for a precision of ±0.1 ppm or better in the stratosphere; it can also measure up to 50000 ppm water vapor, higher than the maximum in the tropical MBL.  Similar to the previous instrument, the large dynamic range was achieved by using two optical paths in the cell, strong and weak absorption lines, and different measurement techniques as described in Sect. 3.  The data were found to have minimal or at least much less sensitivity to instrument temperature compared to the earlier version.  For calibrations up to 200 ppm, we used ultra-pure air and gravimetrically prepared standards [Brewer et al., 2020].  We also calibrated the instrument over the full range of water vapor mixing ratios and pressures found in the troposphere and lower stratosphere (near 0 to ~30000 ppm and ~100-1000 mbar) with a bubbler and a frost point hygrometer (MBW, model 373LX).  Air from the bubbler (or the standards) was passed through the TDL cell and then to the MBW, and also to both instruments in parallel.  Illustrative plots and in-flight comparisons for both instruments are shown in Fig. 11.  Calibrations of the TDL are ongoing, as disagreements with the DLH instrument were observed in ATom, particularly at very high water vapor mixing ratios, >20000 ppm, where the TDL data seem to be anomalously high (not shown).

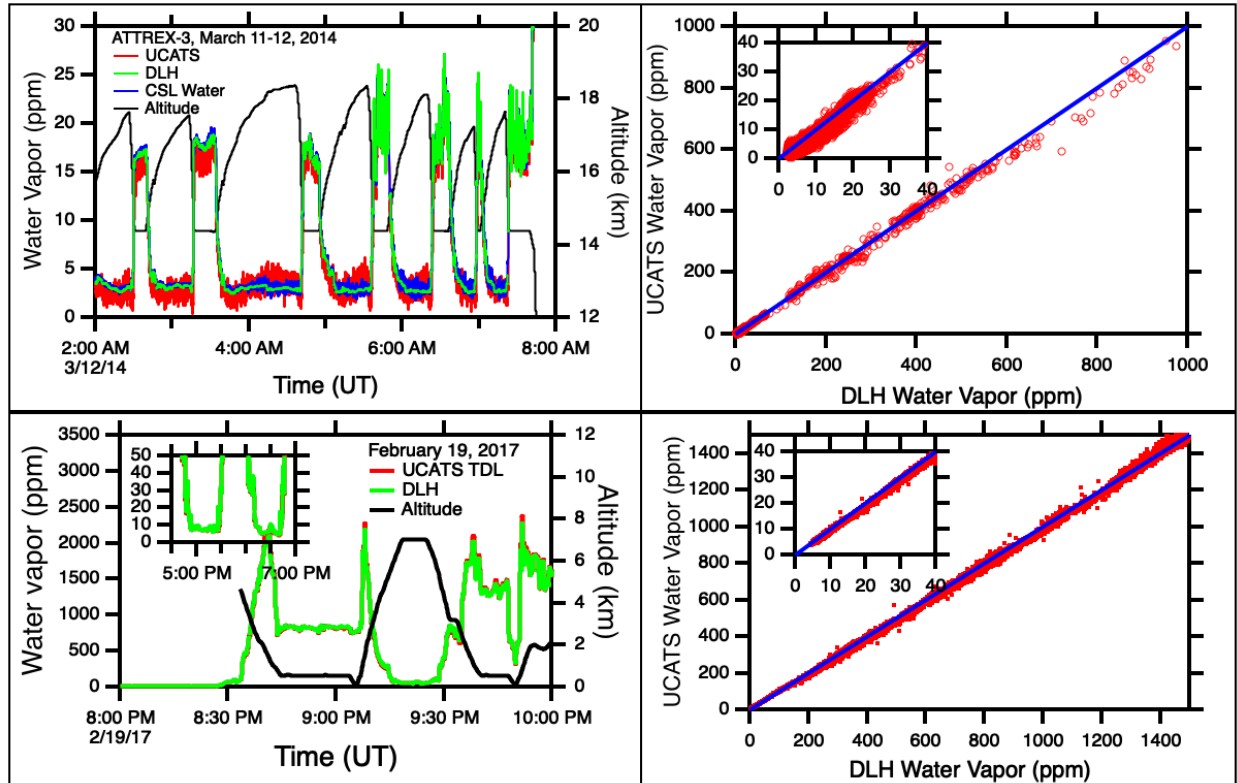

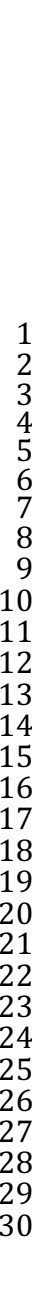

Figure 11: (top) Time series and scatter plots for UCATS TDL water, DLH, and NOAA Water (CSL) during ATTREX. The blue lines in the right hand plots are the 1:1 line for UCATS and DLH for the same flights as on the left. (bottom) Time series of the new UCATS TDL hygrometer and DLH, showing the improved precision at low water vapor mixing ratios. The right hand panel shows the good general agreement between UCATS and DLH over the low and middle range of mixing ratios for the same flight.

## 5 Summary and future plans

The UCATS instrument integrates three different sensor systems into one compact package, for use on UAS and piloted aircraft. The combination of ozone, water vapor, and long-lived trace gases allows for stand-alone experiments with few if any other instruments. It can also contribute to missions on large platforms, by measuring selected (and different) long-lived tracers by GC/ECD, and also providing back-up measurements of species such as ozone and water. UCATS relied on technology developed for the ACATS instrument [Elkins et al., 1996], and improvements to chromatography for the LACE balloon instrument [Moore et al., 2003]. Further improvements made for UCATS were largely to mitigate the effects of water vapor on the GC and ozone systems, allowing measurements throughout the troposphere and stratosphere, as well as continued upgrades of all components.

Over a decade and a half, UCATS has successfully provided trace gas measurements on different types of aircraft for atmospheric science missions with diverse sets of objectives. After demonstration projects focused on the stratosphere, it was used for studies of stratosphere-troposphere exchange and atmospheric transport (START-08 and HIPPO), entry of air into the tropical stratosphere and TTL composition (ATTREX), and tropospheric chemistry (ATom). Over the course of these projects, UCATS evolved from a stratospheric instrument for small UAS payloads to an important contributor on multi-instrument campaigns and geographically extensive tropospheric missions. Table 2 summarizes the data quality over several missions, reflecting improvements in stability for measurements of $N_2O$ and $SF_6$, and improvements in precision and accuracy for ozone. Data from UCATS have been used to help constrain the interhemispheric difference in OH radical concentrations [Patra et al., 2014] and quantify tropospheric age of air and transport using $SF_6$

[Waugh et al., 2013]. Studies have also been performed with UCATS data to probe the composition and structure of the TTL, including halogen chemistry [Jensen et al., 2017; Werner et al., 2017; Navarro et al., 2017], and hydroxyl radical reactivity in the remote troposphere [Thames et al., 2020].

At present, UCATS continues to evolve and has now been upgraded and repackaged for the NASA Dynamics and Chemistry of the Summer Stratosphere (DCOTSS) mission. The initial impetus for this change came from the fact that the UCATS enclosure, extended to accommodate the new ozone and water instruments for ATom, no longer fits in the vertical space available in the upper Q-Bay of the ER-2 aircraft, which will be used for DCOTSS. Since the focus of DCOTSS is on stratospheric ozone and halogen chemistry, the GC is configured to measure $N_2O$, $SF_6$, and CFCs (as initially flown on Altair in 2005), and a third GC channel has been added to measure shorter-lived chlorocarbons including chloroform ($CHCl_3$), carbon tetrachloride ($CCl_4$), and trichloroethylene ($C_2HCl_3$). These changes will allow us to measure much of the organic chlorine budget including the major long-lived organic chlorine compounds, and a few of the more short-lived ones. The repackaged UCATS also has a cleaner and more rational layout, with modular and removeable GC channels, ozone, and water instruments, and more modern electronics, flow controllers, and pressure controllers. Test and science flights on the NASA ER-2 aircraft are now underway in 2021.

**Author contribution**

EJH maintained and operated the UCATS instrument since May 2009, analyzed the data presented here, and wrote the manuscript, FLM designed and built UCATS, operated it on several missions, and worked closely with it throughout, DFH operated UCATS on its first UAS missions and helped to prepare this manuscript, GS Dutton, BDH, JDN, SAM, LPW, AM, and JWE assisted with laboratory work and field missions, JWE was the principle investigator for UCATS for all NASA and NOAA missions through 2020, BRM, SAM, EGH, AFJ, AWR, TDT, LAW, CRT, JP, IB, TBR, BCD, YGR, RC, JVP, SCW, EK, GS Diskin, and TPB provided data and helped with other aspects of the field missions. All the coauthors provided extensive comments and information for this manuscript.

**Competing interests**

The authors declare that they have no conflict of interest.

**Acknowledgements**

This work would not have been possible without the pilots, crew, and staff of the various aircraft involved, the NCAR Research Aircraft Facility, and the NASA Earth Science Project Office. Support was provided for HIPPO by NSF Award #AGS-0628452, for ATTREX by the NASA Earth Venture program, Award #NNA11AA55I, for ATom by NASA Award NNH17AE26I, and additional support was provided by the NASA Upper Atmosphere Research Program, Award # NNH13AV69I. This work was also supported in part by the NOAA Cooperative Agreement with CIRES, NA17OAR4320101. JWE, BDH, and SAM were supported by NOAA's Climate Program Office (CPO). We are grateful to Ru-Shan Gao for discussions, technical expertise, and other support. We thank Andrew Weinheimer, Denise Montzka, and Lisa Kaser for ozone data from the GV during ATTREX and CONTRAST, Craig Williford and Andrew Turnipseed for help with 2B ozone instruments, Randy May for many helpful discussions about TDL water instruments, and Mathew Gentry for thoughtful comments on this manuscript.

**Data availability**

HIPPO data are publicly available at https://www.eol.ucar.edu/field_projects/hippo; GloPac and ATTREX data are available at https://espoarchive.nasa.gov/archive/browse/glopac and /attrex, and ATom data are available at https://daac.ornl.gov/ATOM/campaign/.

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

Appendix A to:

# UAS Chromatograph for Atmospheric Trace Species (UCATS) – a versatile instrument for trace gas measurements on airborne platforms

Eric J. Hintsa et al.

**Table of contents:**

Further details about the UCATS instrument and results from different campaigns are collected here. They include schematics of the ozone and water components, additional data and comparison figures, and details about precision and agreement with other measurements.

**1 Ozone and water instrument schematics**

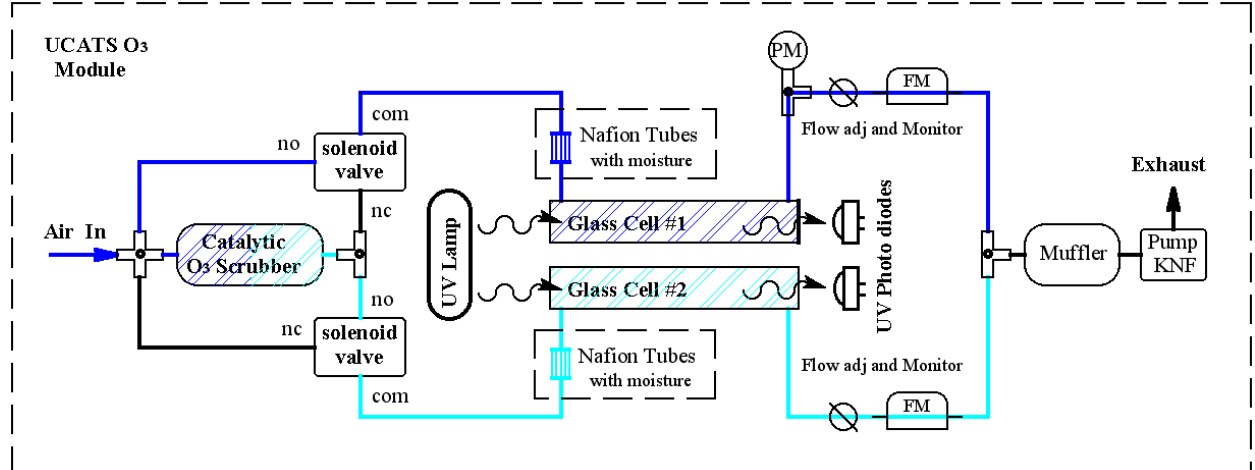

**Figure A1a: For all 2B ozone sensors in UCATS, ambient air is split into two paths, one of which is catalytically**
**scrubbed of ozone with $MnO_2$-coated screens, then alternately sent to the two cells on a 2-second cycle ("no" =**
**normally open, "nc" = normally closed, "com" = common).  The other cell is flushed with ambient air, and a**
**measurement is made every 2 seconds.  Data are averaged to 10 seconds or output at the original 2-second rate.**
**For the Model 211, we humidified the air flow prior to entering the cells with Nafion moisture exchangers.  Cell**
**pressure is measured at the outlet of one of the cells ("PM"), and on the Model 211, flows are measured ("FM")**
**and can be manually adjusted upstream of the pump.  A small settling volume can also be used upstream of the**
**pump to minimize pressure fluctuations.**

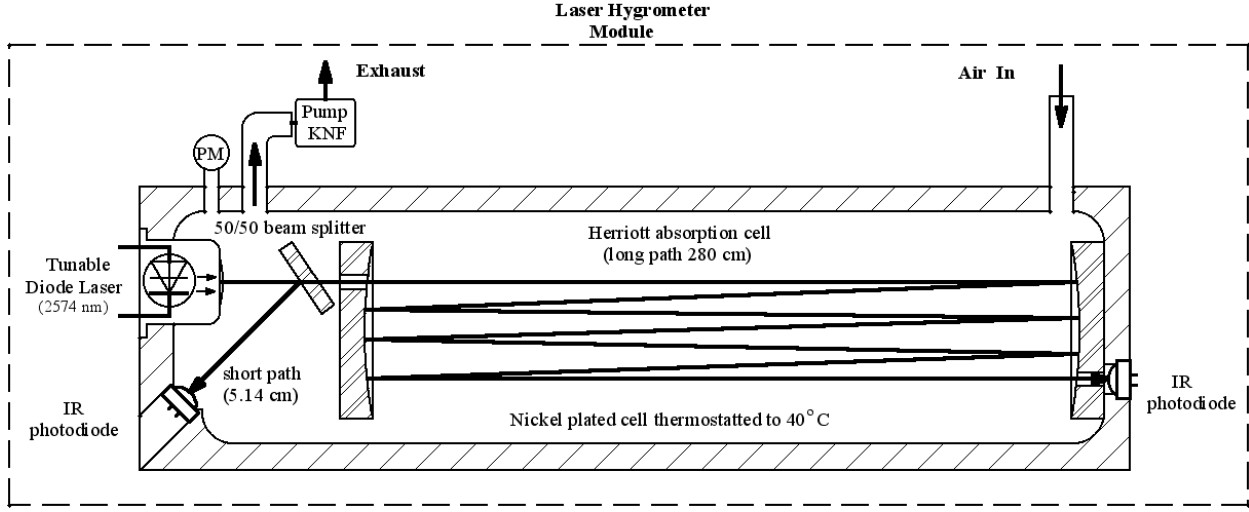

**Figure A1b: The absorption cell for the Port City TDL hygrometer has the diode laser at one end, with a partially**
**reflecting mirror located approximately 2.5 cm from the laser and radiation reflected back to a sealed**
**photodiode for the short path (high water vapor) measurement.  The remainder of the beam is directed by**
**mirrors through a Herriot cell arrangement to a diode opposite the laser for the long-path measurement.  Air**
**flows into the cell through the port farther away from the laser and pressure is measured right at the outlet of the**
**cell to avoid trapped air volumes and to promote smooth flow through the cell.  The electronics are in a separate**
**box (not shown).  In the earlier Maycomm version, the laser was together with the electronics and reached the**
**cell through fiber optic cables, but the cell geometry was similar overall, except that the pressure sensor was in**
the middle of the cell, which could lead to issues with trapped air and delays in the cell drying out after
transitions between wet and dry air masses.
**2 GC Calibrations**

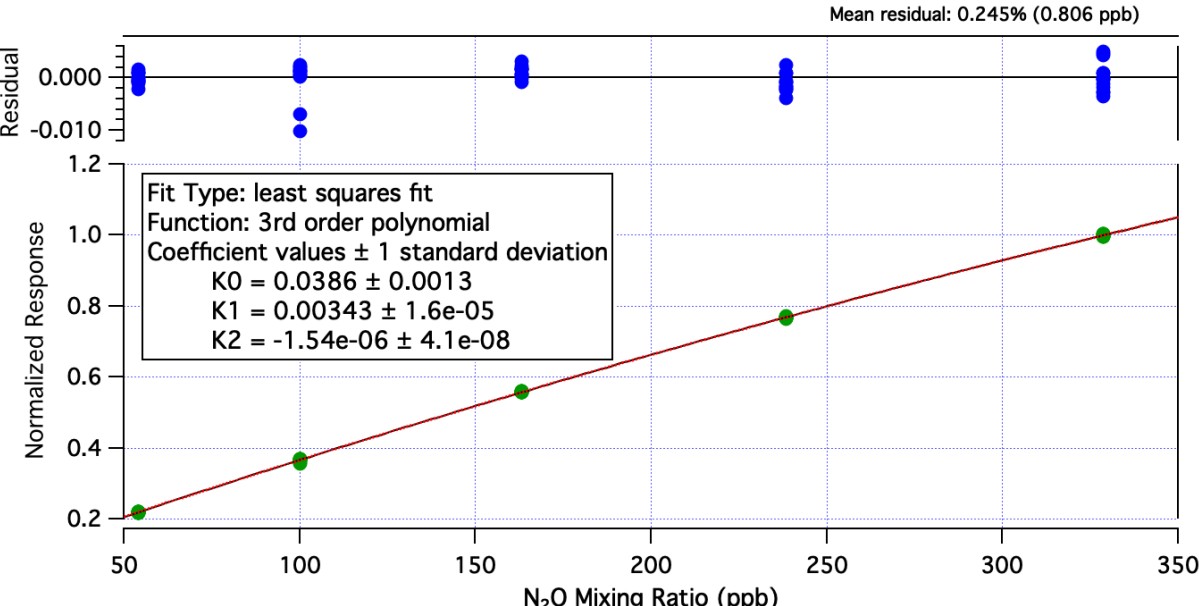

Figure A2: $N_2O$ calibration curve from laboratory experiments during the ATom-2 deployment. UCATS is
calibrated on the ground with a set of standards with precisely measured mixing ratios for all gases. The
calibration gas bottle on the aircraft is filled with air from the flight standard, which is generally background
tropospheric air from Niwot Ridge, CO. In this figure, it is the set of points near 330 ppb. The slight nonlinearity
is taken into account when calculating mixing ratios from measured ambient air and calibration gas samples in
flight. Calibration curves for other molecules measured by UCATS are even closer to linear. The residuals from
the fitted curve and calibration data are shown at the top of the figure and are used in calculating the uncertainty
of reported data.
**3 Tropical N$_2$O and SF$_6$ data**

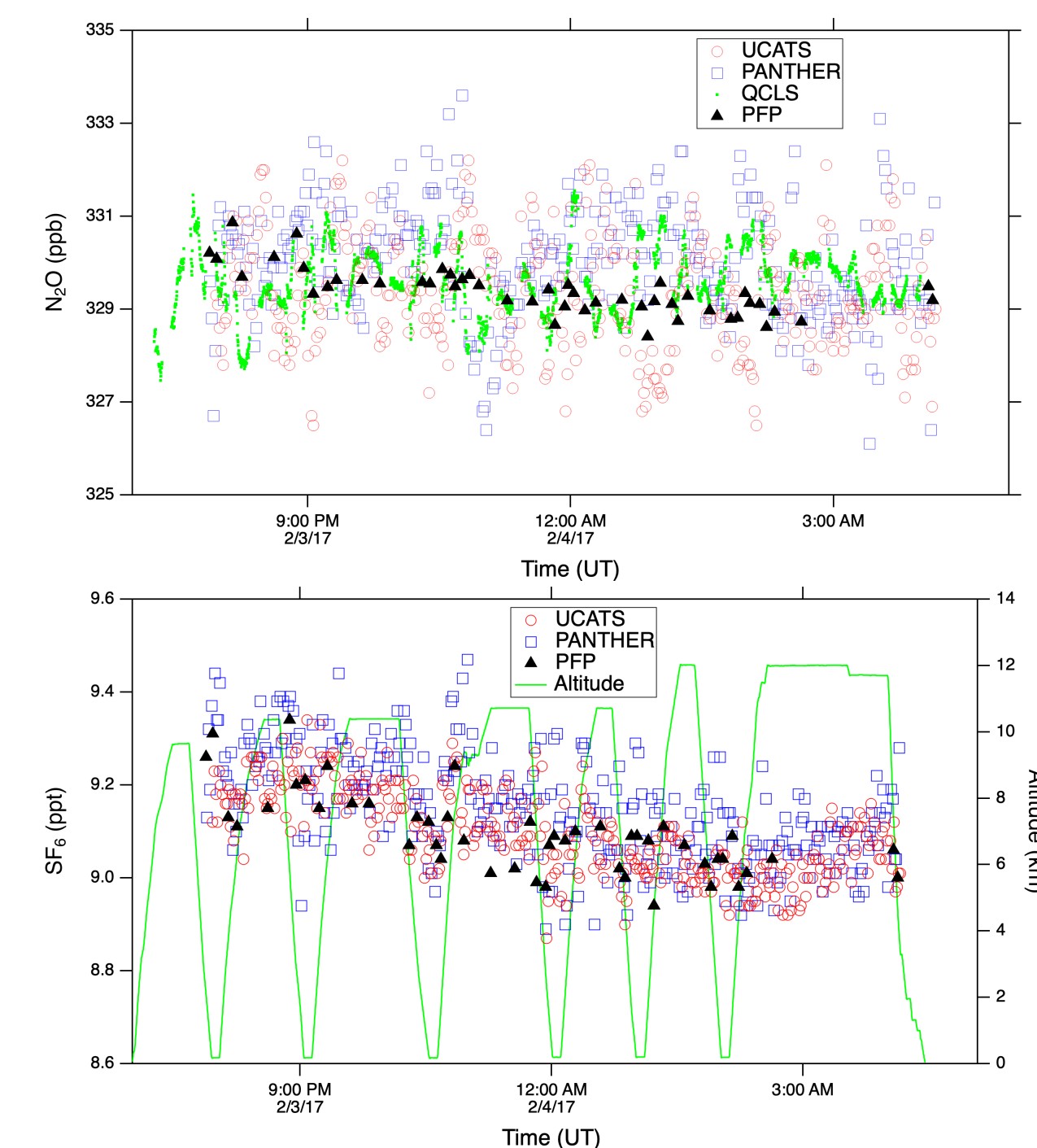

**Figure A3: N$_2$O (top) and SF$_6$ (bottom) time series plots from the February 3, 2017 ATom-2 flight from Kona, HI to**
**Fiji through the tropics. The precision of the *in situ* instruments was near 1 ppb N$_2$O and 0.05 ppt SF$_6$.**
**Throughout this flight the DC-8 remained in the troposphere. The weak vertical gradients in these long-lived**
**tracers (compare to Fig. 5 at high latitudes) allows the latitudinal gradient to be easily discerned (lower values in**
**the Southern Hemisphere), particularly for SF$_6$. Flights through very humid air tested the ability of UCATS to**
**maintain stable chromatography and good precision.**
**4 GloPac ozone**

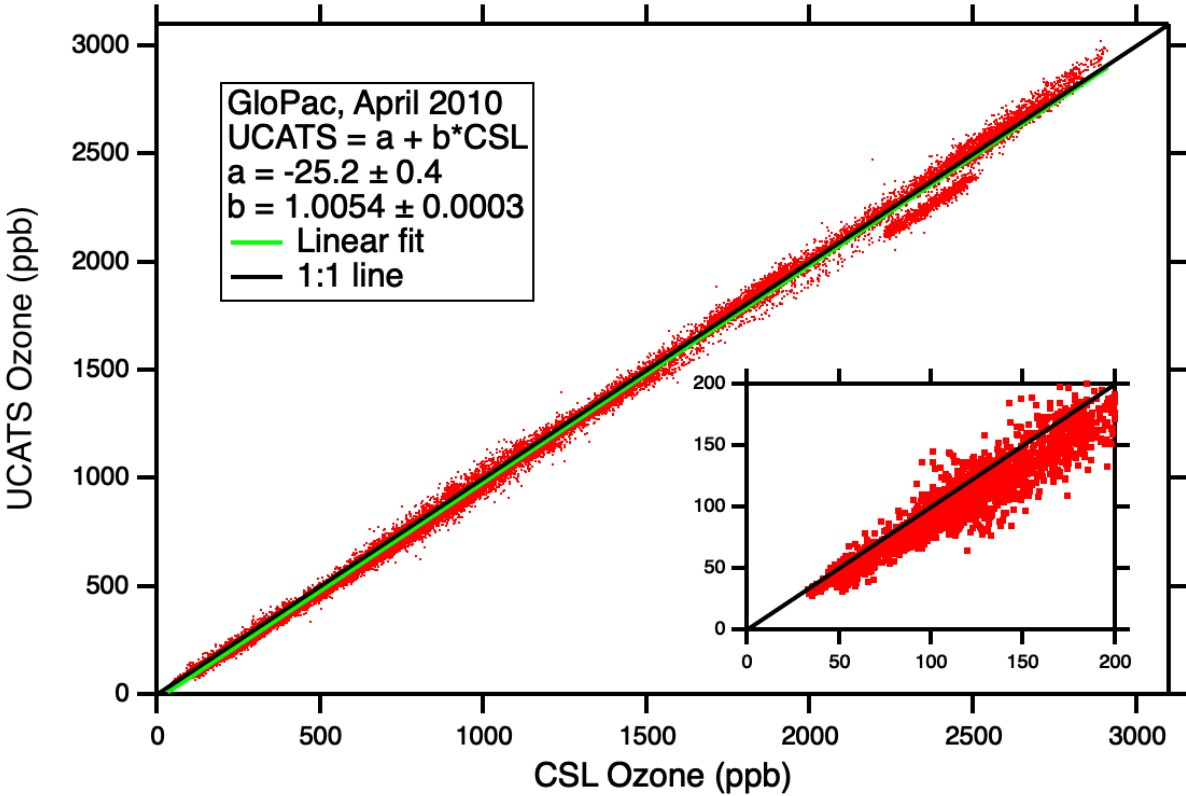

Figure A4: Comparison of UCATS (Model 205) ozone against NOAA CSL Classic Ozone for the entire set of GloPac
flights.  With its long range and duration, including a 28-hour flight on April 23, 2010, the Global Hawk sampled a
wide range of air masses from the tropics to the North Pole, including a polar vortex fragment.  It encountered
very high ozone values considering its maximum altitude is just over 19 km; this was possible during and after
the season of strong descent near the northern hemisphere polar vortex.  Almost all the data shown here were
obtained in the stratosphere; no attempt was made to exclude data from the troposphere on ascents, descents,
and vertical profiles in the tropics.  The linear fit line is always below the 1:1 line in this plot, but UCATS data
were actually about 0-1% higher than Classic Ozone at the highest values, and UCATS data were about 4.5 ppb
lower at the lowest ozone (30-40 ppb, see inset); the linear fit parameters do not quite capture the complete
range of the observations.  The group of points below the 1:1 line near 2500 ppb are almost all from the flight of
April 7; it is not understood what caused this.
**5 ATTREX ozone consistency between different platforms**

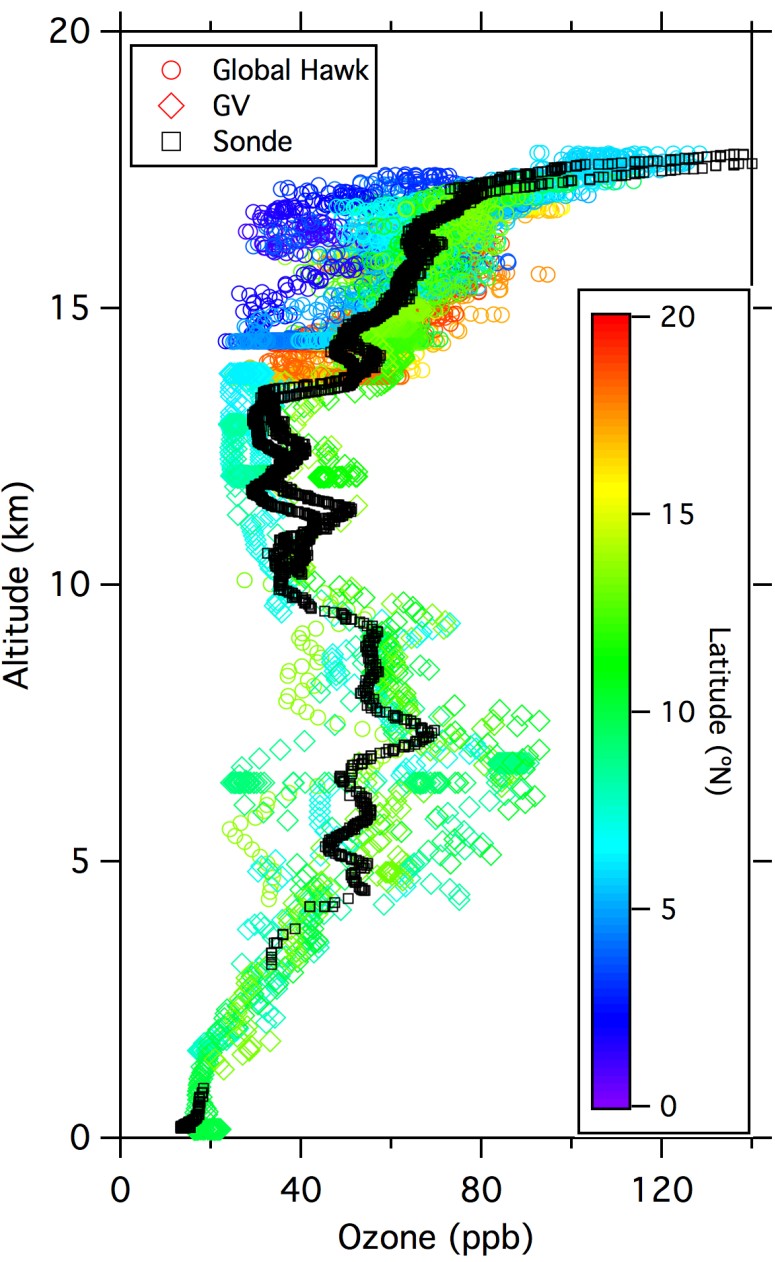

Figure A5: Combined ozone data from the Global Hawk (UCATS) on February 12-13 (circles; mainly above 12 km),
the GV (NCAR) on February 12 (diamonds; 0-14 km), and the ozonesonde (NOAA GML) launch on February 13
(black squares).  The aircraft data are color-coded by latitude; the sonde data (both ascent and descent) are all
from near Guam (13.5°N) and would appear light green if color-coded.  These were not coincident measurements,
as the two aircraft sampled different air masses and the sonde was launched one day later than the GV flight, but
the two-day comparison shows consistency between the various measurements and many of the typical features
encountered in February 2014 over the western tropical Pacific.  Ozone was low at the surface (~20 ppb), with
large variability in the mid-troposphere (Pan et al., 2015, Anderson et al., 2016) caused by frequent encounters
with filaments of high ozone air over a much lower background.  A second minimum is visible in the upper
troposphere up to the base of the TTL.  In the TTL, ozone gradually increased with increasing altitude, with large-
scale variations related to latitude and long-range transport, then increased much more sharply near the top of
the TTL (~17.5 km or 380 K), as air with greater stratospheric character was sampled.
**6 Ozone gradients and precision**

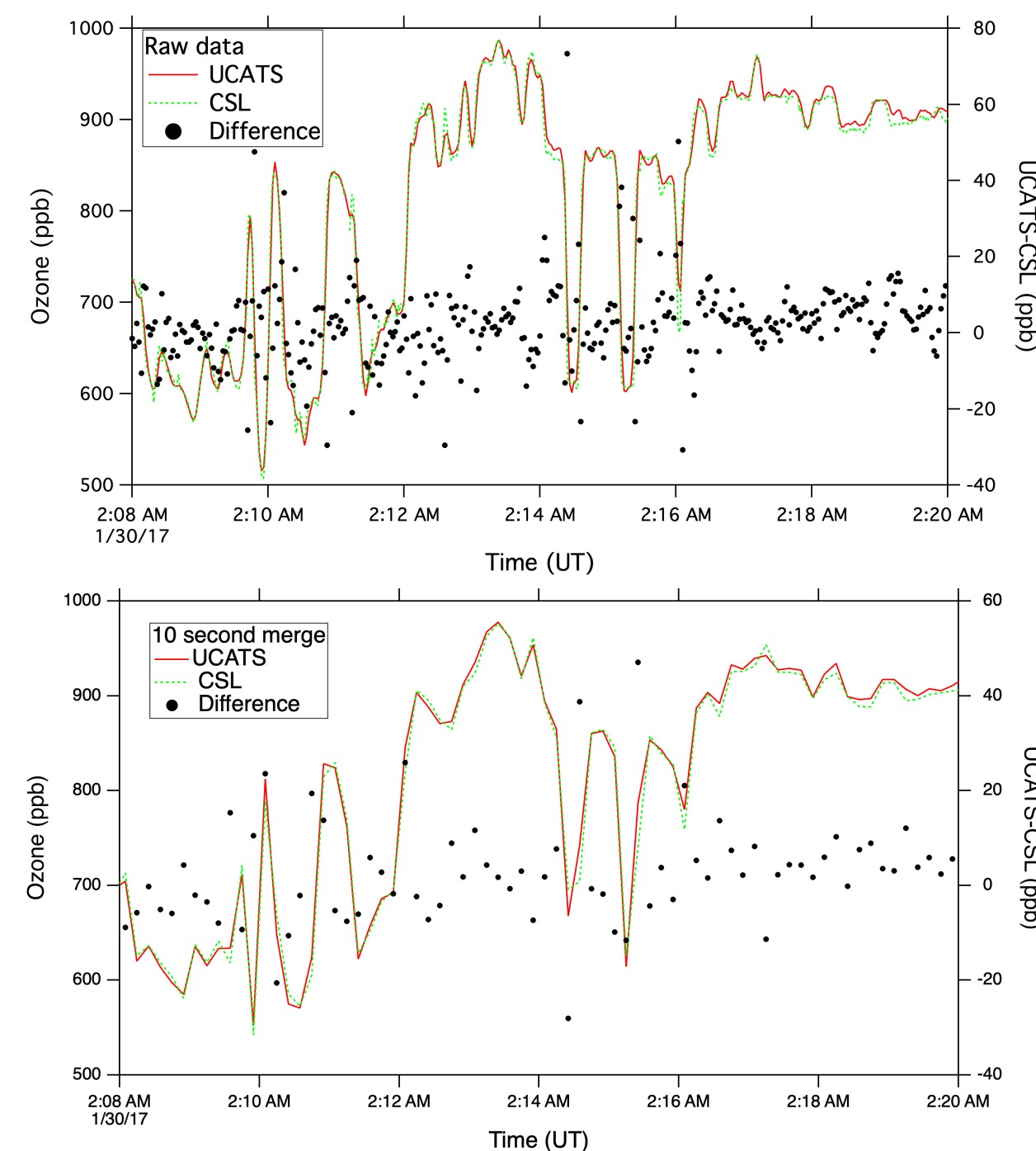

**Figure A6: Archived UCATS and CSL ozone data for the ATom flight of January 29, 2017, from Palmdale, CA to**
**Anchorage, AK (top). The black circles indicate the difference between the two instruments, with 1-second CSL**
**data interpolated to match the sampling times of the 2-second UCATS data. Other than adjustments for timing, no**
**corrections were made to either data set. The visual agreement for the time series is impressive, with both**
**instruments capturing the atmospheric variability, but nonetheless the black circles indicate differences over 40**
**ppb in places. As noted in the main text, this is due to slight offsets in timing, and also from the fact that the 2B**
**instrument has a short "dead time" when flows switch between cells, whereas the CSL data are essentially**
continuous, and reported at 1 Hz for ease of use and to achieve good signal-to-noise.  Where ozone is varying
rapidly, such as this level flight leg at 11.3 km, or ascents and descents, these two effects lead to many of the
"outliers" in comparison plots (such as the ATom panel in Fig. 10).  The lower plot shows the same data, but from
the 10-second merge file commonly used for analyses.  Some of the fine structure in the raw data is washed out,
but there are still differences of up to 5%, even though the two data sets appear to match almost perfectly.  As an
example of the precision possible with the UCATS instrument, we next show a segment near the ocean surface
with much less variability (Figure A7).

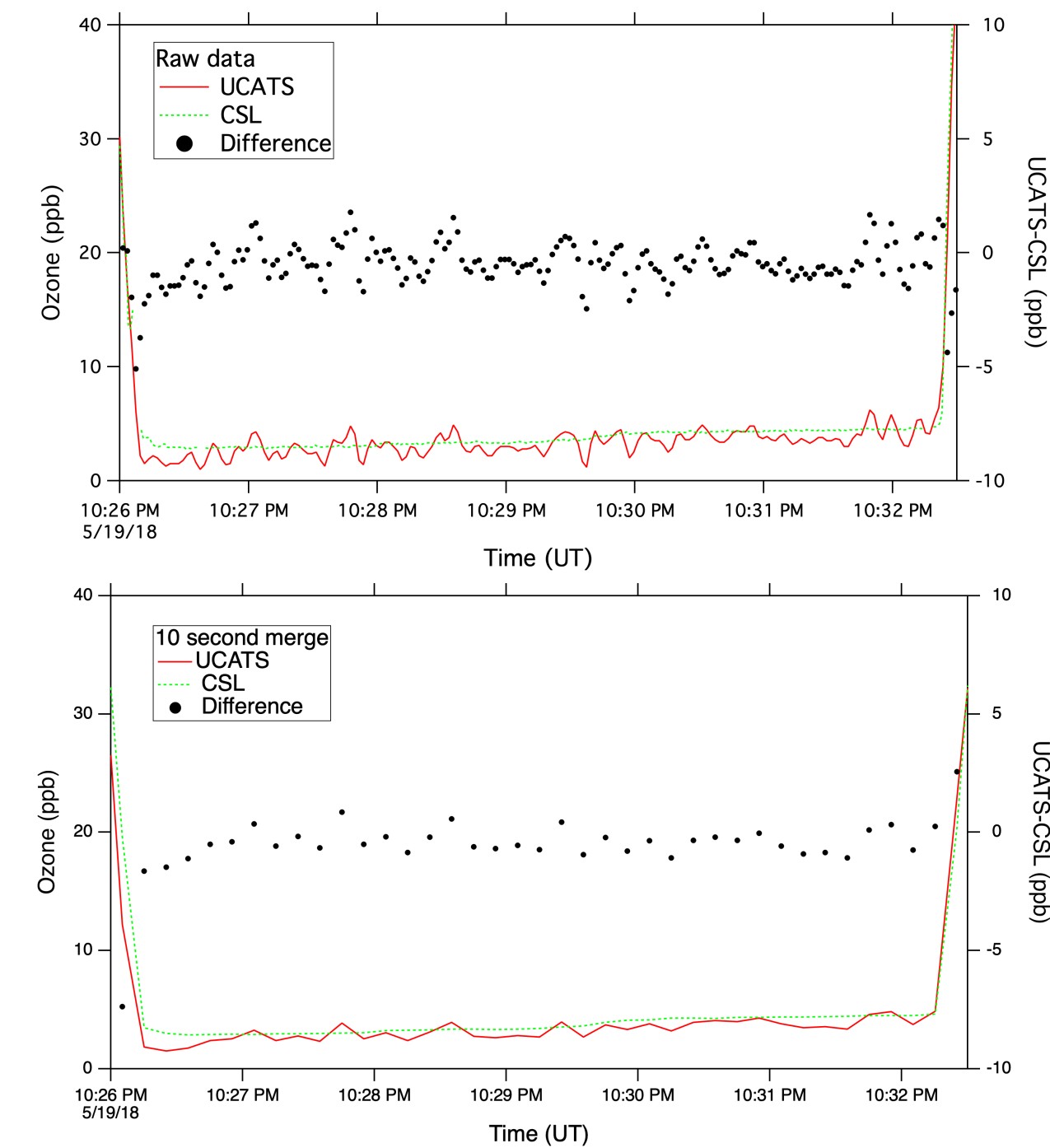

Figure A7: Same as the previous figure, but for May 19, 2018.  Here, at the bottom of a profile over the Beaufort
Sea, ozone was low and nearly constant for about 6 minutes.  The mean difference between data from the two
instruments was 0.4 ppb, and the standard deviation of the difference was 0.5 ppb for the 10-second merge and
**0.7 ppb for 2-second data.  Near the start of plots, the agreement is actually better for the 2-second data than the**
**10-second merge, and at the end of the time series, the "disagreement" between the two instruments changes**
**sign from the 10-second merge to the raw data.  This points to the difficulties of data merges in general, and**
**fundamental limits on comparing two relatively fast data sets.  For ATom, where the goal was to provide**
**comprehensive data sets covering broad regions of the global remote atmosphere, this is insignificant.**
