# Peer review of "UAS Chromatograph for Atmospheric Trace Species"

_Atmospheric Measurement Techniques, 2020_

## Referee Comment (RC2)

**Review on AMTD manuscript by Eric Hintsa et al. "UAS Chromatograph for Atmospheric Trace Species (UCATS) – a versatile instrument for trace gas measurements on airborne platforms"**

**Overall appraisal and general comments:**

UCATS is a fantastic airborne instrument that has produced valuable and high quality scientific data. Its development and numerous deployments over the past 15 years have been a great success story, and the manuscript by Eric Hintsa et al. is interesting and very well written. I absolutely enjoyed reading it.

However, when considering the manuscript as potential paper in Atmospheric Measurement Techniques (AMT), I am having trouble finding clear *purpose* and *added value* for the scientific community. I find the take home message that UCATS "successfully provided trace gas measurements on different types of aircraft for atmospheric science missions" a bit thin, and I would like to see something in terms of "new science" (which I think is there but not very visible): what's unique and novel about the UCATS instrument compared to similar airborne instruments or compared to what has already been presented in earlier publications? Are there any particularly innovative features or solutions to common problems that haven't been adequately solved before? In the conclusions, there is a sentence listing the "challenges encountered", but one or two paragraphs summarizing "innovative solutions found" would be better.

Currently, the manuscript comes across as a hybrid between a descriptive instrument paper, a review article on the work of the research group, and an instrument inter-comparison study. The title suggests that your intention is the former, and I suggest to make the instrument description more comprehensive, and to give more details on innovative aspects, i.e. problem solutions that have not been found or published previously. At the same time, try to shorten or compress the sometimes lengthy "history" bits, and to some extent also the instrument comparisons (concentrate on those analyses that provide valuable information on data quality and homogeneity). Maybe a little bit of restructuring would help to better see and find (i.e. if you search for it) some of the useful pieces of information, e.g. the integration of three different instruments (GC, $O_3$ and $H_2O$ spectrometers) into one compact package, or how you dealt with moisture issues with both the GC part and the $O_3$ sensors. Here, I think dedicated subsections (with subheadings!) would be more valuable than the bits and pieces currently spread throughout the text.

In short, make the instrument description more comprehensive. Shift the focus to and provide more details on *innovations*, where appropriate put them into context of what other groups have done, and clearly list these highlights in the conclusions. This would tremendously increase the added value of your paper and I would comfortably recommend publication in AMT.

**Specific comments:**

Page 2, Line 5: Here in the abstract, you could be a bit more specific on the $O_3$ and $H_2O$ instruments by adding the information that both are commercial sensors measuring by absorption spectroscopy.

Introduction (general):

While the second paragraph on the background and history of the HATS group and the UCATS instrument is certainly interesting, the information given is to some extent redundant with the more detailed and more technical development history given in Section 3. Maybe that part can be shortened here in the introduction with reference to Section 3, or alternatively, you give Section 3 a more technical/topical structure and focus and move the *time line view* to the introduction entirely.

Preceding this, a few lines could be added on why airborne gas chromatography is the preferred method to measure certain trace species and what the advantages and limitations are in general terms. Some references to at least the most important other airborne GC groups/instruments may also be appropriate.

Page 2, Line 32: The statement here is actually not correct, at least with respect to higher altitudes: to my knowledge, the ceiling altitude of the piloted ER-2 is at least equal to that of the unmanned Global Hawk.

Section 2 (general):

The statement in the introduction (page 2, line 55) suggests that the instrument description in Section 2 exclusively refers to the original design of the first version. Is this really the case, and does the block diagram in Figure 1 really reflect this first design? The caption and references made in the text are not always clear in that respect.

In any case, I think it would be most useful to attempt to describe the *latest/best version* in Section 2, in particular with respect to the modular layout. Please try to consider the following:

- It would be interesting to learn in what way the sub-instruments (up to three GC channels, $O_3$ and $H_2O$ spectrometers) efficiently share the space of a common housing, a common power supply and a common PC architecture combining the data in real time. In other words: what are the advantages of packaging everything together in one instrument box rather than flying these instruments separately? How much weight/space/power-draw is saved?

- Optimized weights and dimensions should already be given in the context of the statement on "compact design for small payload spaces" (page 3, line 8), because people's interpretation of what "compact" and "small payload space" actually mean may vary. Maybe mission specific information with respect to box size and weights, and whether gas bottles and $O_3$ and $H_2O$ sensors were integrated or placed outside the box can be summarized in a table that can be referenced here and in other places where appropriate.

- Provide detailed information on what gas bottles are used and at what pressure. Do they need to be refilled for each flight?

Page 5, line 1: By "a series of …", do you mean more than one sensor in one setup, or one sensor for each different instrument version?

Page 5, lines 19 – 28: I'd like to see more detailed information here, i.e.

- What is the overall weight of the entire package with everything that is needed to measure? How has this changed over time?

- Please provide more information on the power system: how much power is needed for the different parts (GCs, $O_3$ and $H_2O$ sensors)? Do you use any converters, filters, etc.? Are there any EMC issues you had to deal with?

- Does Ampro Computer actively control the instruments or is it just used for data acquisition? Are the data from the GCs and the spectroscopic sensors synchronized and merged into one file? Ideally, the main features of the instrument software would be briefly described and the code be made available via a repository.

Page 6, lines 25 – 31: I find the description of what you did and how successful it was a bit sparse and hard to fully understand. Maybe this can be expanded?

Page 7, lines 1 – 7: I see the advantage of flying two sensors to increase time resolution from 10 s to 5 s, but I don't how the addition of the second 2B would substantially remedy the *precision at 100 hPa issue* and bring precision to anywhere near 1 ppb. Please explain this in more detail and convincingly.

Page 7, lines 25 – 31: The described solution to remove moisture that can be applied at large pressure differentials could be interesting to other groups facing similar problems. Consider going into even more detail here, possibly including a drawing so the air flow can be more easily visualized. If you have results from controlled experiments with air of varying moisture, it would be great if you could show them to demonstrate how much moisture can be removed this way, and how it quantitatively solves the problems.

Page 7, lines 33 – 48: It would be good to see some results from your experiments to ensure the accuracy of the $H_2O$ sensors. Also, it would be good to see how consistent the short path absorption / long path absorption / long path 2nd harmonic detection are in the mixing ratio ranges where they overlap.

Page 7, lines 54/55: As the next UCATS deployment is with the ER-2, and the caption to Table 1 suggest that another repackaging is currently being carried out, it might be an idea to add one or two sentences here.

Page 8, lines 12 – 14, 45 – 46 and Figures 2 + 3: I'm not convinced the use of the 1997 ACATS data qualifies as *comparison*. The improvement in precision from ACATS to UCATS seems trivial, and I'm not sure the correlation plots are the best way to demonstrate let alone quantify this improvement. Nevertheless, the demonstration of consistency between the instruments flown in different time periods may be useful for people working with these data to address long term records and trends. Such a consistency demonstration would, of course, become even more useful if more data from the many UCATS missions were included. According to Table 1, $N_2O$ and $SF_6$ were always measured, and $CH_4$ was measured always but in the first Altair campaign. And from the text I read that stratospheric air was sampled at least for some flight sections in many of these campaigns. Besides these general considerations, it would be good to provide a more detailed rationalization including references for the tropospheric growth of $N_2O$, $SF_6$ and $CH_4$.

Page 10, lines 8 – 21: Determining precision using atmospheric measurements of "near-constant $N_2O$" is not ideal because you make assumptions about natural variability. You are trying to work around that by using concurrent data from other instruments (mainly the QCLS) to filter out this natural variability, but you're only replacing the natural variability assumption by an assumption on QCLS precision. Why did you not measure UCATS (including as much of the sampling system as possible) precision in a dedicated experiment repeatedly measuring a standard? To me, that would seem the most intuitive and straightforward method. Alternatively (or additionally) it could be interesting to show the variability over all in-flight CAL gas measurements. If you inject that every 9 minutes, you should have quite good statistics over each mission.

Page 10, lines 21 – 27: Here you discuss accuracy, which I regard as the more relevant parameter when it comes to instrument comparisons. I find it a bit disturbing that the slope of the fit between QCLS and UCATS $N_2O$ being so substantially different from 1.0 is not discussed in more detail. Looking at Figure 4b, there is a high bias of UCATS compared to QCLS at $N_2O$ mixing ratios in the high 200s (stratospheric/mixed/polar vortex outflow?) of 2 – 4 ppb. Depending on the science you do with the data, this can be significant! Even if you haven't been able to fully resolve the reasons for this bias, a detailed discussion and possibly a

recommendation for users of the HIPPO dataset seems warranted. Interestingly, looking at the top right panel of Figure 5, the bias in the same mixing ratio range during ATom appears to be reversed, i.e. now UCATS is lower than QCLS.

Figures 4, 5 and 6: I think it would make sense to use the same symbol for UCATS data in all three comparison figures.

Figure 5: How useful is the lower right panel plotting UCATS/PANTHER/PFP SF6 against QCLS $N_2O$? In my opinion, a plot of UCATS vs. PANTHER $SF_6$ would be much more useful here.

Page 12, lines 25 – 26, Table A1: For people using UCATS data in scientific studies (whom I regard as at least one key audience of this paper), Table A1 is probably one of the most useful pieces of information in the paper. Thus, I first suggest moving this table from the appendix to the main body. And it could also be a good idea to use this table as the basis for the discussion of the instrument evolution in terms of the quality metrics *precision* and *accuracy*. The caption of Table A1 states that the NOAA airborne instruments use the same standards and scales as the surface network. It would be good to include more information on the laboratory and in-flight calibration procedures using these standards (currently, the CAL bottle is shown in Figure 1, and it is stated on page 5, line 21 that this calibration gas is injected every 9 minutes). Most importantly, show results from the calibrations as the first measures for precision and accuracy, and only then discuss comparison to other instruments. Finally, a simple number of 1 ppb for agreement of UCATS $N_2O$ with other instruments for HIPPO is obviously not valid over the entire measurement range (cf. my previous comment). Please check all numbers in the table carefully and don't be too simplistic and/or two optimistic with them!

Page 13, line 15: It is not clear to me why Fig 3A is moved to the Appendix, other than for example Fig 7. Any figure that is discussed in detail in the paper should be shown within the main paper and not in an appendix or supplement.

Page 13, line 26: As an $O_3$ standard is not something you typically get in a bottle, please describe this NIST standard or at least provide an appropriate reference.

Page 15, lines 13 – 15: I don't buy the conclusion of moisture retained in the scrubber explaining the UCATS high bias, at least not based on the evidence presented. There is no doubt that the red line in the top panel of Fig 9 falls above the green line on the ascent sections, but to my eye, it does even more so on the descent sections, when the aircraft moves from drier to wetter air masses.

Page 19, lines 23 – 27: This list of important and valuable science papers using UCATS data could be merged into the introduction. Strictly, they are not summarizing the results of this particular paper.

Page 20, lines 8 – 10: This sounds as if at least the design of this layout is already available. As this new design seems to be a major or even some kind of *final* step in the instrument development (of course, such instruments never stop evolving), it would be good to actually include it in this paper (see my comment on Sections 2 and 3 above).

Appendix A (general):

The layout and content of this *Appendix* appear more like a typical AMT *Supplement*. I suggest moving any Figures and Tables that are really additional information to a supplement and drop the appendix. But as indicated in several comments above, a number of items in the current Appendix probably belong to the main paper and should be moved there.

And I'm not sure about the strategy to leave the description of the $O_3$ and $H_2O$ sensors more or less in the main text but move any (or almost all) illustrative material to the Appendix. Either move the Figures also to the main text, or, alternatively, move the entire description of these commercial sensors into an Appendix (in this case, an Appendix would indeed be appropriate).

Figure A3: What's the peculiar group of points just below the 1:1 line at ~2500 ppb $O_3$? It does not really fall out of the range, but it visibly sticks out so at least a short sentence on this *feature* seems warranted.

Page 19, lines 6 – 9: If you include a supplement, then why not show a figure on this very nice comparison that you mention also?

---

## Author Comment (AC1)

We thank the two reviewers for their careful reading of the manuscript and their comments. I have tried to respond in a way that both addresses the comments and improves the paper. In particular, the reviews suggest emphasizing what is innovative, important and/or significant, making the focus of the paper clearer, and eliminating redundant or unimportant parts. Since that has always been our goal, we are in fundamental agreement. That is described more completely below. There are also some smaller points where either the manuscript just needed a slight clarification or a mistake corrected, and a few where I disagree with the comments and have attempted to explain why.

Anonymous referee #1

The paper by Hintsa et al. attempts to present the NOAA UCATS instrument and discuss its measurement quality. The instrument was deployed on a series of missions over the past 15 years and has evolved considerably during this period. This latter fact presents the main problem I have with the paper. It does not describe one state of the instrument to a degree where this could be seen as a documentation or possibly as a reference for other instrument developers. Rather the paper describes in partly qualitative form the evolution of the instrument over time, partly in a narrative manner. As this evolution spans 15 years, it is virtually impossible to document everything with the desired accuracy. My suggestion would, however, be to add more detail to the description of the most recent set-up. In particular I would like to see more quantitative description of some of the GC parameters (column lengths, flow rates, temperatures) and also chromatograms of the GC channels. In particular I missed information on detector nonlinearity for the ECD channels and also on in-flight calibration frequency and in-flight precision vs. laboratory precision. If the authors estimate all this to be too much detail, I would also be happy to see this as supplementary material. I also suggest that table A1 should be moved to the main part of the publication, whereas the remaining figures in the Appendix could also be moved to a supplement. Next to these more general comment I have a couple of more specific comments given below. Overall the paper fits well into the scope of AMT and I believe it should be published once these points have been addressed.

The reviewer makes a good point here. I tried to emphasize the most recent set-up, but that obviously did not come through well enough, or was masked by other details. We have rewritten sections of the manuscript to more fully describe the instrument as flown in the ATom mission, which is the most recent one for which we have data to present. We added the details suggested by the reviewer about GC parameters, including detector non-linearity (and an associated figure). Many of these details also apply to earlier missions, though there were often small adjustments to the chromatography over time. These changes are mostly in the main text, with some details of more specialized interest in the Appendix. There is still some description of the earlier versions, as well as future changes, but hopefully these are contained and not a distraction. (One exception is for ATTREX ozone data, where we hope to present the scientific analysis of TTL structure and composition in a future publication; a thorough discussion of data quality for this is included.)

p. 3. l. 14-18. I suggest to include some references to ECD detector doping here.

Yes; we added a reference on doping that specifically discusses the interaction of N2O with CO2, CH4, and H2. In UCATS, that also allows us to detect CO (with N2O doping).

p. 3l. 35: is there a model Nr. for the Maycomm hygrometer?

The hygrometers were both custom built, and that is now noted in the text.

p.5. l. 34.: as such papers will be around for a longer time period, I suggest not to use term like "upcoming", as this will be outdated in a short time. Rather use planned for the year X. Similar issues are found at other places in the manuscript. (e.g. p.19., l. 29, p. 20., l. 10)).

Agreed. The original text was written as the extent and impact of the Covid-19 pandemic were just becoming fully realized, and it was unclear how this would affect our schedules. Dates are now inserted, and at present seem likely to be correct.

p.8. l. 48.: I find this "trend-correction" very problematic. While this may be o.k. for SF6 (rather constant trend, very small chemical loss), it is not appropriate for N2O, where due to chemical loss, such a "trend-correction" should be done using a percentage increase. In both cases, this is very crude and it should be clear that some differences may remain which are due to trends.

Yes, part of the reason to make this adjustment was to allow both data sets to be plotted on the same scale, with similar axis ranges. The basic processes of horizontal and vertical transport and photochemistry in the atmosphere that lead to the correlation are the same (though their magnitudes may have changed); other than that there is no profound scientific insight in this figure - it is just a way of easily viewing the two data sets. The increase in SF6 emissions may be the larger source of the differences; N2O has increased much more steadily (and slowly). And you are right that a percentage increase is more appropriate for N2O. I went back and recalculated the scaled POLARIS data with a percentage change for N2O and CH4. The figures changed, and have been updated. Obviously I confused (or unintentionally misled) the reviewers here; the text has been changed to make it clear that that the POLARIS data are mainly for reference to show the similar precision as for ACATS (but faster data rate), as well as the similar (not intended to be identical) structure in the atmosphere.

p. 10. l. 22: and Fig 4: It seems to me that the QCLS data are systematically higher in the troposphere. Have you checked absolute calibration scales? Please also discuss possible ECD non-linearity as mentioned above. This might be able to explain the significant deviations at low N2O values.

Yes, the QCLS data are systematically higher in the troposphere. An offset of approximately 1.2 ppb was systematically observed in the ATom and HIPPO campaigns between QCLS and PFP data. These offsets in QCLS data relative to PFP (the reference instrument on board in both missions) are attributed to the calibration procedure. In a new paper (now cited in this manuscript), Gonzalez et al. [2021] adjust QCLS data for the small differences in the troposphere compared to the PFP samples in ATom data; this reduces QCLS N2O values by a little more than 1 ppb, and the figures with ATom data in this manuscript (Figures 5 and A3) have also been

changed. The HIPPO data have now also been corrected in a similar way. We account for ECD non-linearity in our calibrations, though we do rely on that non-linearity remaining constant in flight (and from flight to flight). We think that one possibility for the deviations at low N2O is from the fact that the in-flight standards used with the QCLS in HIPPO typically spanned a range from ~300 ppb to 330-350 ppb; the focus was on obtaining the best data in the troposphere, and (low) stratospheric values of N2O were obtained by extrapolation. The vast majority of the QCLS data in HIPPO is within the span of the high and low calibration gases, though the stratospheric data stand out in the figures because of their large range. We adjusted the text to try to make this clearer.

p. 12. l. 17 and Figure 5: The correlation between N2O and SF6 looks quite remarkable. Could the authors add some more discussion on this? Are these data from different hemispheres?

Yes, there is a lot of information in this plot. In the stratosphere (or stratospherically influenced air), older air has lower N2O (from photochemical loss, and the slightly smaller N2O mixing ratio in older air) and lower SF6 (tropospheric SF6 was lower at earlier times), leading to the positive correlation shown. (The oldest air is in the lower left, as in Figure 2.) Data from both hemispheres (~70 S to 80 N) are shown in the plot, and the data near 330 ppb N2O (actually most of the data points) are from the troposphere. This part of the plot reflects the very strong latitudinal gradient in SF6, with lower SF6 in the southern hemisphere and higher SF6 in the northern hemisphere. This leads to a much steeper apparent slope (N2O also has a latitudinal gradient, but weaker, also with lower values in the SH). Transitions between the troposphere and the stratosphere lead to mixing lines between the two branches, with the lower SF6 transitions (8.6-9.0 ppt) in the SH, the higher SF6 (>9.5 ppt) in the NH, and the tropical transition near 9.1 ppt. Part of this has been added to the text.

---

## Author Comment (AC2)

We thank the two reviewers for their careful reading of the manuscript and their comments. I have tried to respond in a way that both addresses the comments and improves the paper. In particular, the reviews suggest emphasizing what is innovative, important and/or significant, making the focus of the paper clearer, and eliminating redundant or unimportant parts. Since that has always been our goal, we are in fundamental agreement. That is described more completely below. There are also some smaller points where either the manuscript just needed a slight clarification or a mistake corrected, and a few where I disagree with the comments and have attempted to explain why.

Referee #2 (Marc von Hobe)

**Overall appraisal and general comments:**
UCATS is a fantastic airborne instrument that has produced valuable and high quality scientific data. Its development and numerous deployments over the past 15 years have been a great success story, and the manuscript by Eric Hintsa et al. is interesting and very well written. I absolutely enjoyed reading it.
However, when considering the manuscript as potential paper in Atmospheric Measurement Techniques (AMT), I am having trouble finding clear *purpose* and *added value* for the scientific community. I find the take home message that UCATS "successfully provided trace gas measurements on different types of aircraft for atmospheric science missions" a bit thin, and I would like to see something in terms of "new science" (which I think is there but not very visible): what's unique and novel about the UCATS instrument compared to similar airborne instruments or compared to what has already been presented in earlier publications? Are there any particularly innovative features or solutions to common problems that haven't been adequately solved before? In the conclusions, there is a sentence listing the "challenges encountered", but one or two paragraphs summarizing "innovative solutions found" would be better.
Currently, the manuscript comes across as a hybrid between a descriptive instrument paper, a review article on the work of the research group, and an instrument inter-comparison study. The title suggests that your intention is the former, and I suggest to make the instrument description more comprehensive, and to give more details on innovative aspects, i.e. problem solutions that have not been found or published previously. At the same time, try to shorten or compress the sometimes lengthy "history" bits, and to some extent also the instrument comparisons (concentrate on those analyses that provide valuable information on data quality and homogeneity). Maybe a little bit of restructuring would help to better see and find (i.e. if you search for it) some of the useful pieces of information, e.g. the integration of three different instruments (GC, O3 and H2O spectrometers) into one compact package, or how you dealt with moisture issues with both the GC part and the O3 sensors. Here, I think dedicated subsections (with subheadings!) would be more valuable than the bits and pieces currently spread throughout the text.

Thank you for your overall appraisal of the UCATS instrument! We intended this as a descriptive instrument paper, to describe UCATS for all to see, and so that science publications could cite this paper. Each publication could, with a few sentences or less, describe the

particular measurements used and any special conditions, and quickly move on to the scientific results. As more detail was added to this manuscript, the intention may have become less clear. We used the reviewer's comments above to restructure the paper. In particular, I tried to compress the history parts, and also removed a few repetitive sections that should have been dealt with earlier. I did leave much of the instrument intercomparison studies, because I think they are one of the best tests of how well the instrument performed.

We added a paragraph in the last section about the technical accomplishments of UCATS. These are largely the integration of the three systems into one instrument, the efforts to deal with moisture issues such that quality data can be obtained in the troposphere, the stratosphere, and in transitions between the two, and to some extent the effort to adapt small, inexpensive ozone sensors for high quality measurements from aircraft. These are the main accomplishments of UCATS, though there are some other incremental improvements mentioned in the paper. Responding to Reviewer #1's main comments and the comments here, we made the instrument description more comprehensive. I also added subheadings to subsections and tried to do a better job of putting the bits and pieces in the appropriate subsection (and including them only once).

In short, make the instrument description more comprehensive. Shift the focus to and provide more details on *innovations*, where appropriate put them into context of what other groups have done, and clearly list these highlights in the conclusions. This would tremendously increase the added value of your paper and I would comfortably recommend publication in AMT.

Yes, in addition to the changes described above, we also added references to other airborne gas chromatography instruments.

**Specific comments:**

Page 2, Line 5: Here in the abstract, you could be a bit more specific on the O3 and H2O instruments by adding the information that both are commercial sensors measuring by absorption spectroscopy.

Yes, I added that. The 2B is more of an "off-the-shelf" commercial instrument (at least until we took it apart to fit it into UCATS, and modified it), while the TDL hygrometers are custom built right from the start; this should be clear from the respective sections in the main text.

Introduction (general):
While the second paragraph on the background and history of the HATS group and the UCATS instrument is certainly interesting, the information given is to some extent redundant with the more detailed and more technical development history given in Section 3. Maybe that part can be shortened here in the introduction with reference to Section 3, or alternatively, you give Section 3 a more technical/topical structure and focus and move the *time line view* to the introduction entirely.

We shortened the history part of the introduction, and tried to include only what is needed in Section 3. I hope we have removed all the redundancy here and elsewhere in the paper.

Preceding this, a few lines could be added on why airborne gas chromatography is the preferred method to measure certain trace species and what the advantages and limitations are in general terms. Some references to at least the most important other airborne GC groups/instruments may also be appropriate.

We added a couple of sentences on why GC/ECD is the preferred method to detect certain trace species - sensitivity, selectivity, and a time response approaching 1 minute, as well as references to other airborne GC instruments. The main drawback is that GCs prefer to be operated continuously, and their performance improves with longer operating times. This is a challenge in an aircraft environment, where power is often shut off, and is cycled frequently, along with fluctuations in T, P, regular mechanical vibration, and the occasional bit of turbulence.

Page 2, Line 32: The statement here is actually not correct, at least with respect to higher altitudes: to my knowledge, the ceiling altitude of the piloted ER-2 is at least equal to that of the unmanned Global Hawk.

Yes, you are correct. UAS have shown great promise and potential for high altitude flight, but to our knowledge, no UAS (including the Global Hawk) with the ability to support any kind of sensor payload has flown higher than the ER-2, whose top altitude is around 21 km. The text has been fixed to reflect this.

Section 2 (general):
The statement in the introduction (page 2, line 55) suggests that the instrument description in Section 2 exclusively refers to the original design of the first version. Is this really the case, and does the block diagram in Figure 1 really reflect this first design? The caption and references made in the text are not always clear in that respect.

We have changed Figure 1 to reflect the configuration in ATom. We changed the text in several places to make this clear as well.

In any case, I think it would be most useful to attempt to describe the *latest/best version* in Section 2, in particular with respect to the modular layout. Please try to consider the following:
- It would be interesting to learn in what way the sub-instruments (up to three GC channels, O3 and H2O spectrometers) efficiently share the space of a common housing, a common power supply and a common PC architecture combining the data in real time. In other words: what are the advantages of packaging everything together in one instrument box rather than flying these instruments separately? How much weight/space/power-draw is saved?

Both reviewers agree broadly on this point, and we are emphasizing the version flown in ATom, which is the latest version for which we have flight data. Almost all the weight of UCATS is from the GCs. The water vapor TDL weighs about 5 lbs, much of it from the absorption cell, and ozone is about 1 lb. Power draws are also heavily weighted toward the GC, and the water and ozone sensors were mostly just squeezed into leftover space (on the sides and top) from the GC. However, this is from the perspective that UCATS is a 2-channel GC, with ozone and water

added on.  If water and ozone were flown separately, they would each need their own housing, power supply/ies, data system, and water vapor would need its own large pump as well, or a larger and heavier inlet to provide ram pressure and flow.  Given that all these were required for the GC system, water and ozone could be added with comparatively small additional weight and current draw, and almost no extra space (until the upgraded systems for ATom required increasing the height of the enclosure in order to fit them inside).

- Optimized weights and dimensions should already be given in the context of the statement on "compact design for small payload spaces" (page 3, line 8), because people's interpretation of what "compact" and "small payload space" actually mean may vary. Maybe mission specific information with respect to box size and weights, and whether gas bottles and O3 and H2O sensors were integrated or placed outside the box can be summarized in a table that can be referenced here and in other places where appropriate.

We removed the language about "compact design for small payload spaces" from the start of section 2, and rewrote this paragraph in the context of the other changes to the manuscript. Information about size and weights is in what is now subsection 2.5 (Physical characteristics), which follows the sections on each component of UCATS.  I think it makes more sense at the end of section 2.

- Provide detailed information on what gas bottles are used and at what pressure. Do they need to be refilled for each flight?

That is now in the text.  The N2 bottle typically needs to be refilled every one or two flights, depending on flight duration, the calibration air bottle is filled once per deployment, and the small dopant bottles last for over a year without refilling.

Page 5, line 1: By "a series of …", do you mean more than one sensor in one setup, or one sensor for each different instrument version?

This has been rewritten to make it clearer; thank you for pointing out the ambiguity here.  We have used two different models of ozone sensors in UCATS, and integrated them in different configurations.  I think the description is now more straightforward.

Page 5, lines 19 – 28: I'd like to see more detailed information here, i.e.
- What is the overall weight of the entire package with everything that is needed to measure? How has this changed over time?
- Please provide more information on the power system: how much power is needed for the different parts (GCs, O3 and H2O sensors)? Do you use any converters, filters, etc.? Are there any EMC issues you had to deal with?
- Does Ampro Computer actively control the instruments or is it just used for data acquisition? Are the data from the GCs and the spectroscopic sensors synchronized and merged into one file? Ideally, the main features of the instrument software would be briefly described and the code be made available via a repository.

All this information is now added to section 2.  By carefully twisting wire bundles (and using appropriate gauge wires) we have largely avoided EMC issues, even in a package as dense as UCATS.  The one electrical issue has been electrical noise on the very sensitive signal from the ECD; however this has been found to be almost entirely from the heater on the relevant ECD, and can be alleviated by careful routing of the heater and signal cables inside the ECD can.  The computer controls the GC by actively switching valves, flows, and pressures; the ozone and water instruments are self-contained and only their outputs are recorded.  Both instrument control and data analysis are controlled by a large set of different files.  The code could be archived, but it would difficult to describe them completely enough to be useful to anyone else.

Page 6, lines 25 – 31: I find the description of what you did and how successful it was a bit sparse and hard to fully understand. Maybe this can be expanded?

Yes, this has been expanded.

Page 7, lines 1 – 7: I see the advantage of flying two sensors to increase time resolution from 10 s to 5 s, but I don't how the addition of the second 2B would substantially remedy the *precision at 100 hPa issue* and bring precision to anywhere near 1 ppb. Please explain this in more detail and convincingly.

By averaging twice as much (independent and equally accurate) data together, one can in principle improve the precision.  But as the reviewer points out, what we effectively did was to double the data rate.  The precision can be improved by averaging sections of data together with the same conditions (and with twice as many data points, the signal-to-noise will improve for that section).  This works well on level flight segments, but is less effective on ascents and descents, when conditions are rapidly changing.  For many recent aircraft campaigns, 10-second averages have been created to put all data on a common time stamp.  This has been found to be a good compromise between sensitivity of the measurements, creating a workable data set that is easy to use, and without averaging out much atmospheric variability.  For certain applications (obviously flux measurements, but also aircraft plume crossings, cloud encounters, and other sharp transitions such as shown in Figure A6) the fastest data rate is needed and used.  One other advantage of a second ozone instrument is that UV photometers (and also chemiluminescence instruments) invariably have occasional noise spikes, likely from a bad data read, or a spike in a PMT, or something else.  These cause short, positive or negative excursions in the data and are typically removed by hand.  At the same time, plume crossings or other sharp features can appear very similar; a second ozone instrument makes it easy to see if they are "real".  These events typically occur a few times per flight, and are therefore insignificant in overall statistics (and usually insignificant in the atmosphere as well).  Nonetheless, they show up clearly in scatter plots, and tend to draw the eye because they are outliers.  I don't think all of this should be in the paper, but have tried to explain it here as best I can.  I added one sentence to the manuscript for clarification.

Page 7, lines 25 – 31: The described solution to remove moisture that can be applied at large pressure differentials could be interesting to other groups facing similar problems. Consider going into even more detail here, possibly including a drawing so the air flow can be more easily visualized. If you have results from controlled experiments with air of varying moisture, it would

be great if you could show them to demonstrate how much moisture can be removed this way, and how it quantitatively solves the problems.

Our solution is really very simple; I'm a little disappointed that we didn't come up with it sooner - it would have made the HIPPO data in the tropics a lot better. All that we did was put the "soft" components (Nafion moisture exchangers) into a sealed box, with a connecting gas line to the air flow downstream of the instrument. This kept the pressure on both sides of the Nafion approximately equal, even as ambient (outside) pressure varied from 150 to over 1000 mbar. For the ozone system, we had one tube going into the box with ~100 sccm of moistened cabin air, and another tube connected to a pressure sensor (mostly to test the system initially in case something went drastically wrong - it never did). The remaining components of the flow path (Teflon tubing, connectors, scrubber, cell, etc.) can easily sustain a larger differential pressure; they just need to be leak-free to prevent cabin air from diluting the air samples being measured. It would be worthwhile and interesting to do some quantitative tests to show how much water is added (or removed, in the case of really wet ambient air) but we have not done that. Mostly, once we verified that the solution worked, we moved on to other things (there were plenty of those in the first ATom deployment). Yes, this could be useful to other groups, and I hope that anyone who could benefit from it does so.

Page 7, lines 33 – 48: It would be good to see some results from your experiments to ensure the accuracy of the H2O sensors. Also, it would be good to see how consistent the short path absorption / long path absorption / long path 2nd harmonic detection are in the mixing ratio ranges where they overlap.

We added a figure addressing this in the Appendix.

Page 7, lines 54/55: As the next UCATS deployment is with the ER-2, and the caption to Table 1 suggest that another repackaging is currently being carried out, it might be an idea to add one or two sentences here.

Yes, I added a "pointer" to the section where this is described.

Page 8, lines 12–14, 45–46 and Figures 2 + 3: I'm not convinced the use of the 1997 ACATS data qualifies as *comparison*. The improvement in precision from ACATS to UCATS seems trivial, and I'm not sure the correlation plots are the best way to demonstrate let alone quantify this improvement. Nevertheless, the demonstration of consistency between the instruments flown in different time periods may be useful for people working with these data to address long term records and trends. Such a consistency demonstration would, of course, become even more useful if more data from the many UCATS missions were included. According to Table 1, N2O and SF6 were always measured, and CH4 was measured always but in the first Altair campaign. And from the text I read that stratospheric air was sampled at least for some flight sections in many of these campaigns. Besides these general considerations, it would be good to provide a more detailed rationalization including references for the tropospheric growth of N2O, SF6 and CH4.

This is largely addressed in the response to reviewer #1. The point was to show that the smaller UCATS instrument could achieve the same or better precision as the older ACATS instrument, but with a much faster data rate (that is now added). The text has been changed, and a reference to tropospheric concentrations and growth has now been added.

Page 10, lines 8 – 21: Determining precision using atmospheric measurements of "near-constant N2O" is not ideal because you make assumptions about natural variability. You are trying to work around that by using concurrent data from other instruments (mainly the QCLS) to filter out this natural variability, but you're only replacing the natural variability assumption by an assumption on QCLS precision. Why did you not measure UCATS (including as much of the sampling system as possible) precision in a dedicated experiment repeatedly measuring a standard? To me, that would seem the most intuitive and straightforward method. Alternatively (or additionally) it could be interesting to show the variability over all in-flight CAL gas measurements. If you inject that every 9 minutes, you should have quite good statistics over each mission.

There are many ways to determine (or at least estimate) precision, and I think it is worthwhile to explore as many of them as possible. The reviewer is correct that using atmospheric measurements of near-constant N2O are not ideal because of natural variability. The same can be said for comparisons with the (very precise) QCLS instrument. However, we think they produce an upper bound to precision (that is, precision can only be better), and under actual flight conditions. The reviewer is also correct that we have a large set of in-flight cal gas measurements; these are generally consistent with the derived precision. We are cautious about claiming the same precision as the cal data (at least for HIPPO), because the air samples have variable water vapor whereas the calibration gas is dried when initially collected. For the (always dry) ATTREX and GloPac data and the improved conditions in ATom this is probably a good assumption, and the derived precision is consistent with the other methods. And yes, we have done (and do) many experiments in the laboratory with repetitive injections of a standard. These work really well, and typically yield precisions of about 0.2% or better for all molecules except $H_2$ and CO (about 0.5%). But because of changing temperatures and pressures, power cycling, etc., it is often not possible to achieve this in flight. That is part of the challenge of operating a GC system on aircraft.

Page 10, lines 21 – 27: Here you discuss accuracy, which I regard as the more relevant parameter when it comes to instrument comparisons. I find it a bit disturbing that the slope of the fit between QCLS and UCATS N2O being so substantially different from 1.0 is not discussed in more detail. Looking at Figure 4b, there is a high bias of UCATS compared to QCLS at N2O mixing ratios in the high 200s (stratospheric/mixed/polar vortex outflow?) of 2–4 ppb. Depending on the science you do with the data, this can be significant! Even if you haven't been able to fully resolve the reasons for this bias, a detailed discussion and possibly a recommendation for users of the HIPPO dataset seems warranted. Interestingly, looking at the top right panel of Figure 5, the bias in the same mixing ratio range during ATom appears to be reversed, i.e. now UCATS is lower than QCLS.

The reviewer is correct here. We revised this to try to improve the discussion. This is also discussed in a very recent paper [Gonzalez et al., 2021, Impact of stratospheric air and surface emissions on tropospheric nitrous oxide during ATom, in ACPD] which takes an intensive look at $N_2O$ data and scientific implications based on the ATom data. From that paper, the QCLS instrument paper [Santoni et al., 2014], and discussions with the participants, we tried to produce the best data set possible and describe it. We note that in the Gonzalez et al. paper, the QCLS ATom data have been revised downward slightly, and we are adopting this change for HIPPO data as well. The text here has been modified, and Figures 4 and 5 have been remade to reflect the changes in the data.

Figures 4, 5 and 6: I think it would make sense to use the same symbol for UCATS data in all three comparison figures.

Thank you for noticing and pointing that out; that is now fixed.

Figure 5: How useful is the lower right panel plotting UCATS/PANTHER/PFP SF6 against QCLS N2O? In my opinion, a plot of UCATS vs. PANTHER SF6 would be much more useful here.
Page 12, lines 25 – 26, Table A1: For people using UCATS data in scientific studies (whom I regard as at least one key audience of this paper), Table A1 is probably one of the most useful pieces of information in the paper. Thus, I first suggest moving this table from the appendix to the main body. And it could also be a good idea to use this table as the basis for the discussion of the instrument evolution in terms of the quality metrics *precision* and *accuracy*. The caption of Table A1 states that the NOAA airborne instruments use the same standards and scales as the surface network. It would be good to include more information on the laboratory and in-flight calibration procedures using these standards (currently, the CAL bottle is shown in Figure 1, and it is stated on page 5, line 21 that this calibration gas is injected every 9 minutes). Most importantly, show results from the calibrations as the first measures for precision and accuracy, and only then discuss comparison to other instruments. Finally, a simple number of 1 ppb for agreement of UCATS N2O with other instruments for HIPPO is obviously not valid over the entire measurement range (cf. my previous comment). Please check all numbers in the table carefully and don't be too simplistic and/or two optimistic with them!

Because PANTHER and UCATS measure SF6 and N2O every 70 seconds, and are not generally overlapping (in fact, we would prefer they be staggered every 35 seconds to provide the most data to the mission, though we do not always actively try to do that), it is not possible to plot UCATS data directly vs. PANTHER data from a flight. One can argue whether the typical 10-second data merge is averaging over too much atmospheric variability, but a 70-second average definitely would. That said, I have done this kind of comparison, and it is roughly consistent with what we show here - I just could not defend it in a paper. As suggested by both reviewers, we moved Table A1 to the main text. And I checked and updated a few numbers in the table, and added some clarifying notes. Thank you for pointing that out.

Page 13, line 15: It is not clear to me why Fig 3A is moved to the Appendix, other than for example Fig 7. Any figure that is discussed in detail in the paper should be shown within the main paper and not in an appendix or supplement.

Figure 3A (now Figure A4) is so similar to Figure 7, except that it is for a different mission and there was only one ozone instrument in UCATS at the time. I originally considered not including it at all, but thought it would add somewhat to the presentation. Also, it is not really discussed in detail in the paper, other than at line 15, and its own caption. If this is not the perception of the reviewer, then something else is misleading or not clear in the manuscript.

Page 13, line 26: As an O3 standard is not something you typically get in a bottle, please describe this NIST standard or at least provide an appropriate reference.

Sorry, this was a sloppy choice of wording. Prior to sale and delivery, 2B instruments are calibrated against a reference standard by the company. We use a Thermo Fisher Model 49i Primary Standard to check the calibration before and after each deployment, as briefly described in Section 2. This is a (much larger) commercial instrument, with precision and accuracy suitable for our requirements, and is periodically recertified by NIST. The text has been changed to reflect this, and also that the calibration did not change over the missions that UCATS participated in.

Page 15, lines 13 – 15: I don't buy the conclusion of moisture retained in the scrubber explaining the UCATS high bias, at least not based on the evidence presented. There is no doubt that the red line in the top panel of Fig 9 falls above the green line on the ascent sections, but to my eye, it does even more so on the descent sections, when the aircraft moves from drier to wetter air masses.

You are correct that the disagreements occurred on both ascents and descents. And from flight to flight, they were not always exactly the same. However, they always (and pretty much only) occurred in the tropics when the aircraft transitioned into and out of the boundary layer. And in ATom, the 2B Model 211, with the moisture exchanger, did not experience these issues, while the Model 205 (carried for reference) continued to show them; the Nafion moisture exchangers are promised to transfer water and little or nothing else. We did not do extensive experiments in flight to prove this, but the results are roughly consistent with the more careful experiments of Wilson and Birks [2006]. We adjusted the wording slightly to reflect this; again, sorry for the careless wording in the original.

Page 19, lines 23 – 27: This list of important and valuable science papers using UCATS data could be merged into the introduction. Strictly, they are not summarizing the results of this particular paper.

We think the list of papers is more appropriate after the instrument and data have been described. If it is critical, I can move this into the introduction, but I think it reads better as is.

Page 20, lines 8 – 10: This sounds as if at least the design of this layout is already available. As this new design seems to be a major or even some kind of *final* step in the instrument development (of course, such instruments never stop evolving), it would be good to actually include it in this paper (see my comment on Sections 2 and 3 above).

Yes, the design of the new layout is available now.  However, we decided to focus here even more closely on the configuration for ATom, the most recent one for which we have data.  We think the new design will be even better, but the changes can be described in one paragraph, building off the present manuscript.

Appendix A (general):
The layout and content of this *Appendix* appear more like a typical AMT *Supplement*. I suggest moving any Figures and Tables that are really additional information to a supplement and drop the appendix. But as indicated in several comments above, a number of items in the current Appendix probably belong to the main paper and should be moved there.
And I'm not sure about the strategy to leave the description of the O3 and H2O sensors more or less in the main text but move any (or almost all) illustrative material to the Appendix. Either move the Figures also to the main text, or, alternatively, move the entire description of these commercial sensors into an Appendix (in this case, an Appendix would indeed be appropriate).

I initially wrote what was intended to be a Supplement, and in fact the references to some of the figures were (mistakenly) left as "S3", for example, rather than "A3" in the text (now fixed).  But in preparing the manuscript for review, I read from the AMT guidelines:

*Appendices: all material required to understand the essential aspects of the paper such as experimental methods, data, and interpretation should preferably be included in the main text. Additional figures, tables, as well as technical and theoretical developments which are not critical to support the conclusion of the paper, but which provide extra detail and/or support useful for experts in the field and whose inclusion in the main text would disrupt the flow of descriptions or demonstrations may be presented as appendices. These should be labelled with capital letters: Appendix A, Appendix B etc. Equations, figures and tables should be numbered as (A1), Fig. B5 or Table C6, respectively. Please keep in mind that appendices are part of the manuscript whereas supplements (see below) are published along with the manuscript.*

and:

*If you have **supplementary material** to your manuscript which does not meet the above-mentioned asset criteria to be hosted by a reliable repository, you can submit your [supplement as a \*.zip archive or single \*.pdf file](). The overall file size of such a supplement is limited to 50 MB. Larger supplements have to be submitted to a reliable data repository in any case receiving a DOI, cited in your manuscript, and included in your reference list.*

So, what I consider to be "extra detail and support useful for [readers]" seemingly should be in an Appendix.  On reading this reviewer comment, I was at a loss as to how to respond, so I inquired with the journal.  It seems that a lot of this is left to the discretion of the author (and I guess the reviewers); the only real difference is that Appendices are published immediately after the main text, while Supplements are published separately, with their own DOI.  Since nowadays, almost everyone reads online (or downloads and prints), it doesn't seem to make much difference.  I will clear this with the journal once more, but I think I am doing the right thing. (Though I freely admit that my initial viewpoint was the same as the reviewer's.)  I do

think that the diagrams of the commercial instruments (water and ozone) would disrupt the flow of the paper and just make it even more too long; in the Appendix, a reader can simply skip them (and most will) or read if interested.

Figure A3: What's the peculiar group of points just below the 1:1 line at ~2500 ppb O3? It does not really fall out of the range, but it visibly sticks out so at least a short sentence on this *feature* seems warranted.

These points are all from one flight; it is not clear why the agreement between the two instruments was different on that particular day. I added a sentence to clarify that. We did not see anything like that in ATTREX (Figure 7) for either of the 2B instruments flown.

Page 19, lines 6 – 9: If you include a supplement, then why not show a figure on this very nice comparison that you mention also?

OK, I had already prepared this figure; it is now included with a short interpretive caption.